# Activated endothelial cells induce a distinct type of astrocytic reactivity

Xavier Taylor [1,2,7], Pablo Cisternas[1,2], Nur Jury[1,2], Pablo Martinez[1,2], Xiaoqing Huang[3], Yanwen You[1,2], Javier Redding-Ochoa[4], Ruben Vidal[1,5], Jie Zhang [3,6], Juan Troncoso[4] & Cristian A. Lasagna-Reeves [1,2,6✉]

Reactive astrogliosis is a universal response of astrocytes to abnormal events and injuries. Studies have shown that proinflammatory microglia can polarize astrocytes (designated A1 astrocytes) toward a neurotoxic phenotype characterized by increased Complement Component 3 (C3) expression. It is still unclear if inflammatory stimuli from other cell types may also be capable of inducing a subset of $C3^+$ neurotoxic astrocytes. Here, we show that a subtype of $C3^+$ neurotoxic astrocytes is induced by activated endothelial cells that is distinct from astrocytes activated by microglia. Furthermore, we show that endothelial-induced astrocytes have upregulated expression of A1 astrocytic genes and exhibit a distinctive extracellular matrix remodeling profile. Finally, we demonstrate that endothelial-induced astrocytes are Decorin-positive and are associated with vascular amyloid deposits but not parenchymal amyloid plaques in mouse models and AD/CAA patients. These findings demonstrate the existence of potentially extensive and subtle functional diversity of $C3^+$-reactive astrocytes.

[1] Stark Neurosciences Research Institute, Indiana University School of Medicine, Indianapolis, IN 46202, USA. [2] Department of Anatomy, Cell Biology & Physiology, Indiana University School of Medicine, Indianapolis, IN 46202, USA. [3] Department of Biostatistics and Health Data Science, Indiana University School of Medicine, Indianapolis, IN 46202, USA. [4] Department of Pathology, School of Medicine, Johns Hopkins University, Baltimore, MD, USA. [5] Department of Pathology and Laboratory Medicine, Indiana University School of Medicine, Indianapolis, IN 46202, USA. [6] Center for Computational Biology and Bioinformatics, Indiana University School of Medicine, Indianapolis, IN, USA. [7] Present address: Neuroscience Discovery, Lilly Research Laboratories, Eli Lilly & Co, Lilly Corporate Center, Indianapolis, IN 46225, USA. ✉email: clasagna@iu.edu

Reactive astrogliosis is a universal response of astrocytes to pathological events and injuries, including neurodegenerative disease, trauma, ischemia, and infection; however, their role in these conditions is not fully understood[1]. The ability of astrocytes to regulate the blood-brain barrier (BBB), remodel the extracellular space, control immune cells, and regulate synapse formation and function may all influence how the brain fares during and following injury[2,3]. In response to pathology, astrocytes engage in molecularly defined changes in transcriptional regulation leading to biochemical, morphological, and physiological remodeling, culminating in the gain of new function(s) or the loss or upregulation of homeostatic functions[1].

For decades, the study of neurodegenerative diseases has focused on mechanisms of toxicity and neuronal cell death. The general assumption has been that most of the deleterious events leading to brain dysfunction are essentially sustained by neuronal cell-autonomous molecular cascades. However, several other cell types, such as astrocytes, surround and interact with neurons, playing a major role in maintaining normal neuron function and survival[4,5]. It has recently been shown that activation of microglia and astrocytes might not be independent events[6] and that activated microglia can directly polarize a subset of astrocytes toward a neurotoxic phenotype (designated A1 astrocytes); however, it has become increasingly clear that more complex neuroinflammatory subtypes of astrocytic reactivity exist that do not necessarily align with the recently established A1/A2 dichotomy[1,6,7]. This microglia-induced A1 astrocytic subtype is characterized by increased expression of complement component 3 (C3) and has been identified in human patients and mouse models of Alzheimer's disease (AD), Huntington's disease, amyotrophic lateral sclerosis, multiple sclerosis, Parkinson's disease, prion disease and frontotemporal dementia[6,8–10]. We and others have recently reported that reactive C3$^+$ astrocytes are highly abundant and cluster around vascular amyloid deposits without major microglial reactivity in three different mouse models of Cerebral Amyloid Angiopathy (CAA)[11–13]. The lack of microglial reactivity suggests that proinflammatory signaling from another source may be responsible and sufficient for the induction of reactive astrocytes associated with vascular amyloid deposits. While it is unknown if damaged endothelial cells can polarize astrocytes to a neurotoxic subtype, endothelial cells can produce and release proinflammatory cytokines under toxic stimulation, suggesting that these cells could be capable of inducing astrocyte reactivity[14,15]. Therefore, following a well-established approach for generating microglia-induced A1 reactive astrocytes[6], we aimed to determine if activated endothelial cells could also promote the polarization of astrocytes to a neurotoxic phenotype.

Indeed, our results demonstrate that activated endothelial cells strongly induce the activation of astrocytes by increasing their expression of C3 and other components of the complement system, similar to the microglia-induced A1 astrocytic phenotype. Nevertheless, RNA-seq analysis revealed that astrocytes activated by endothelial cells have different molecular signatures than astrocytes activated by microglia. These endothelial-induced reactive astrocytes have a significant increase in the expression of extracellular matrix genes, such as Decorin (Dcn), compared to astrocytes activated by microglia. Furthermore, these Decorin$^+$ C3$^+$ astrocytes maintain their phagocytic capacity and exhibit a neurotoxic gain of function and are associated with vascular amyloid deposits but not parenchymal amyloid plaques in mouse models and AD/CAA patients.

## Results

**Astrocytes activated by endothelial cells are C3$^+$, neurotoxic and retain phagocytic capacity.** It has been well established that microglia can directly polarize astrocytes toward a neurotoxic

phenotype characterized by increased expression of complement component 3 (C3) as a typical marker[6]. Therefore, we aimed to determine if activated endothelial cells can also polarize astrocytes toward a neurotoxic phenotype. After confirming the purity of each culture (Supplementary Fig. 1a, b), primary microglia and endothelial cultures were stimulated with the proinflammatory stimulus LPS[16]. Control or activated microglia conditioned media (MCM) or endothelial conditioned media (ECM) was then concentrated and evaluated with an endotoxin quantification kit to ensure no endotoxin contamination (Supplementary Fig. 1c). Endotoxin-free media was applied to astrocytes, which were then immunostained for C3 (Fig. 1a, d). C3$^+$ astrocytes were present in both activated MCM and ECM treatment conditions, with a 44% increase in the C3$^+$ area in astrocytes treated with activated MCM compared with that in the control (Fig. 1b, c) and an 86% increase in the C3$^+$ area in astrocytes treated with activated ECM compared with that in the control (Fig. 1e, f). Overall, these observations indicate that activated endothelial cells robustly induce C3$^+$ astrocytes, suggestive of A1-like astrocytic induction. We then evaluated the neurotoxic capacity of astrocytes activated by LPS-treated endothelial cells. Astrocytic conditioned media (ACM) from astrocytes previously activated by LPS-treated endothelial or microglial cells was collected and applied to primary neuronal cultures (Fig. 2a, e). As previously reported[6], we found that the ACM collected from activated MCM-treated astrocytes significantly reduces neuronal viability, as determined through an MTS viability assay (Fig. 2b). To confirm these results, neuronal viability was also assessed through measuring endogenous esterase activity, and we observed a significant reduction in neuronal viability using a Calcein AM viability assay (Fig. 2c, d). Interestingly, we found that ACM collected from activated ECM-treated astrocytes also demonstrated a neurotoxic effect, with a significant reduction in neuronal viability observed with MTS (Fig. 2f) and Calcein AM (Fig. 2g, h) viability assays. Then, we evaluated whether astrocytes previously activated by LPS-treated endothelial or microglial cells induce impairment of the synaptic integrity in viable neurons. We performed IF for the pre- and post-synaptic markers Synapsin-1 and PSD95 and assessed their colocalization. Our results showed a decrease in the total number of clusters positive for Synapsin-1 and PSD95 expression in neurons incubated with ACM collected from activated MCM-treated astrocytes compared with the number of clusters positive for these proteins in the control samples (Fig. 2i–j). Furthermore, we observed a decrease in the number of clusters positive for Synapsin-1 and PSD95 in neurons treated with ACM collected from activated ECM-treated astrocytes (Fig. 2k–l), suggesting that reactive astrocytes activated by endothelial cells also impair synaptic integrity. A novel study has suggested that long-chain saturated fatty acids in lipoparticles may mediate microglia-induced A1 astrocyte neurotoxicity; however, it is clear that reducing these lipids does not completely eliminate neurotoxicity, suggesting that future work is needed to discover other astrocyte-derived toxins[17] and to determine if similar neurotoxic factors are secreted from endothelial-induced reactive astrocytes.

It has been reported that microglia-induced A1 astrocytes lose many of their homeostatic functions, including phagocytic capacity[6]. To investigate the phagocytic capacity of astrocytes, we measured the level of engulfment of zymosan bioparticles by astrocytes treated with activated MCM or ECM (Fig. 2m, p). We found that control MCM-treated astrocytes were able to robustly phagocytose zymosan bioparticles, but upon activated-MCM treatment, this capacity was substantially reduced, with a 53% reduction in net phagocytic effect (Fig. 2n, o). However, we observed no change in the net phagocytic effect of astrocytes treated with activated-ECM compared to that of those treated with control-ECM (Fig. 2q, r). In addition, we have recently

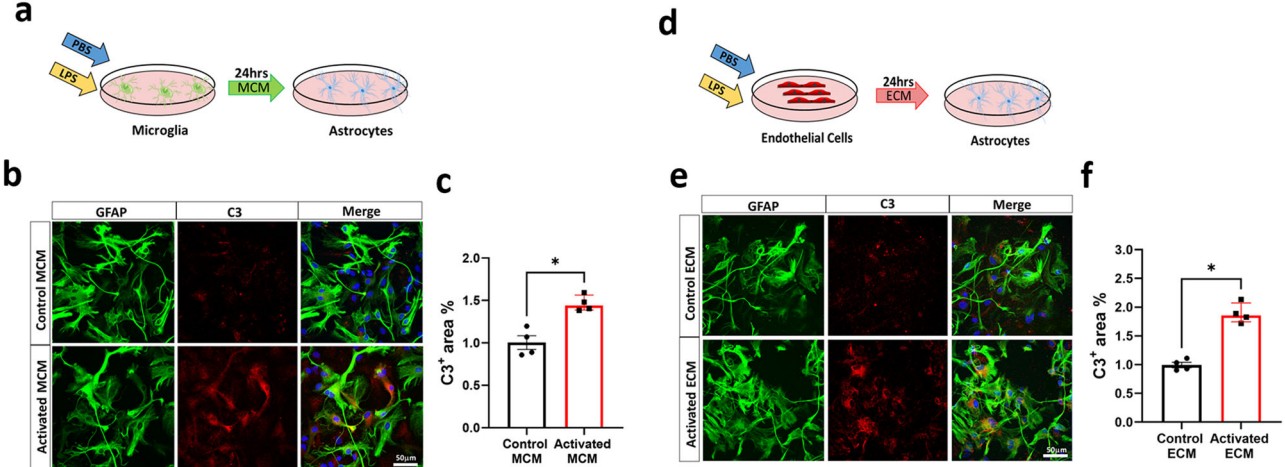

**Fig. 1 C3+ Astrocytes are induced by LPS-Activated ECM. a** Schematic diagram showing treatment of astrocytes with control or activated microglia conditioned medium (MCM) **b** Double immunofluorescence of complement component 3 (C3, red) and astrocytes (GFAP, green) treated with control or activated MCM in primary astrocyte cultures. C3 and GFAP immunoreactivity overlay (merge) **c** Quantification of C3+ area (%) of control or activated MCM treated astrocytes. **d** Schematic diagram showing treatment of astrocytes with control or activated endothelial conditioned medium (ECM) **e** Double immunofluorescence of complement component 3 (C3, red) and astrocytes (GFAP, green) treated with control or activated ECM in primary astrocyte cultures. C3 and GFAP immunoreactivity overlay (merge) **f** Quantification of C3+ area (%) of control or activated ECM treated astrocytes. The number of cells analyzed was 20 per image from 10 different images per culture from 4 independent cultures. Results are shown as the median ± IQR of $n = 4$. Asterisks indicate significant differences, where *$p < 0.05$ by Mann-Whitney test. Scale bar 50 μm.

demonstrated that A1 astrocytes induce a decrease in TREM2 protein levels in microglia, suggesting a feedback loop between astrocytes and microglia[11]. To further characterize astrocytes activated by LPS-treated endothelial cells, we evaluated the effect of ACM on TREM2 levels in microglial culture. The ACM from astrocytes activated by MCM or ECM was collected and applied to primary microglial cultures (Supplementary Fig. 2a, e). As previously reported[11], in primary microglia cultures incubated with ACM from activated MCM treatments, TREM2 protein levels were significantly decreased in comparison with those incubated in ACM from control MCM treatments (Supplementary Fig. 2a–d) Interestingly, in primary microglia cultures incubated with ACM from activated ECM treatments, TREM2 protein levels were also significantly lower than those in the control (Supplementary Fig. 2e–h). Collectively these data indicate that astrocytes activated by LPS-treated microglial or endothelial cells may have overlapping functions in their ability to modulate neuronal survival and microglia homeostasis but differ in their intrinsic phagocytic capacity, suggesting that, while both astrocytes can be identified by increased C3+ expression, the molecular and transcriptional profile of these astrocytes may be distinctly different.

**Endothelial and microglial stimuli produce different reactive astrocyte transcriptomes.** Given the identified heterogeneity of reactive astrocytes between stimuli[18], we wondered whether there would be heterogeneity in astrocytic gene expression in response to activated microglia or endothelial cells. Thus, we performed a partial transcriptomic analysis using a NanoString Technologies glial profiling panel that evaluates the expression of 770 genes, allowing us to perform gene set enrichment analysis (GSEA). GSEA revealed that endothelial-induced astrocytes have overlapping gene sets and pathways with microglia-induced A1 astrocytes (Fig. 3a, d and Supplementary Data 1). Importantly, A1 astrocytic gene sets were the primary scored pathway in astrocytes activated by microglia (Fig. 3a), whereas the complement system pathways and A1 astrocytic gene sets were the top two scored pathways in endothelial activated astrocytes (Fig. 3d),

suggesting that LPS-treated endothelial cells can activate astrocytes into an A1-like genetic profile. In addition, astrocytes treated with activated MCM versus non-activated astrocytes demonstrated increased undirected differential expression of A1 astrocytic genes listed in the NanoString Glia panel (Fig. 3b). Similarly, in astrocytes treated with activated ECM, an increased undirected differential expression of A1 astrocytic genes was observed (Fig. 3e). However, some of the A1 astrocytic genes, such as guanine-binding protein 2 (GBP2), which is known to be highly upregulated in astrocytes activated by microglia ([6,19] and Fig. 3c), were downregulated in astrocytes activated by endothelial cells (Fig. 3f), suggesting that these astrocytes activated by endothelial cells have a genetic profile distinct from that of the previously described A1 astrocytes activated by microglia[6]. The notion that astrocytes activated by endothelial cells are distinct from those activated by microglia is supported by the fact that certain pathways are solely enriched in one kind of activated astrocyte (Fig. 3a, d).

To further determine global transcriptional differences between astrocytes activated by MCM and those activated by ECM, we performed RNA sequencing analysis on astrocytic cultures. Differential expression analysis (fold change > 1.5, $p < 0.05$) revealed a set of genes that are exclusively enriched in astrocyte transcriptomes treated with activated ECM or activated MCM along with genes commonly expressed by both subtypes (Fig. 4a and Supplementary Fig. 3a, b). A total of 130 genes (74 upregulated and 56 downregulated) were uniquely differentially expressed in astrocytes treated with activated ECM and 233 genes (157 upregulated and 76 downregulated) were uniquely differentially expressed in astrocytes treated with MCM. We used DAVID bioinformatics to identify the most significantly upregulated Gene ontology (GO) term pathways. GO enrichment analysis revealed that astrocytes treated with activated ECM have a distinct profile highly enriched in genes corresponding to proteinaceous extracellular matrix, extracellular region, extracellular space, and extracellular exosome. Importantly, gene sets corresponding to extracellular matrix components were the most highly enriched overall (Fig. 4b and Supplementary Data 2). In the case of astrocytes treated with activated MCM, GO enrichment analysis

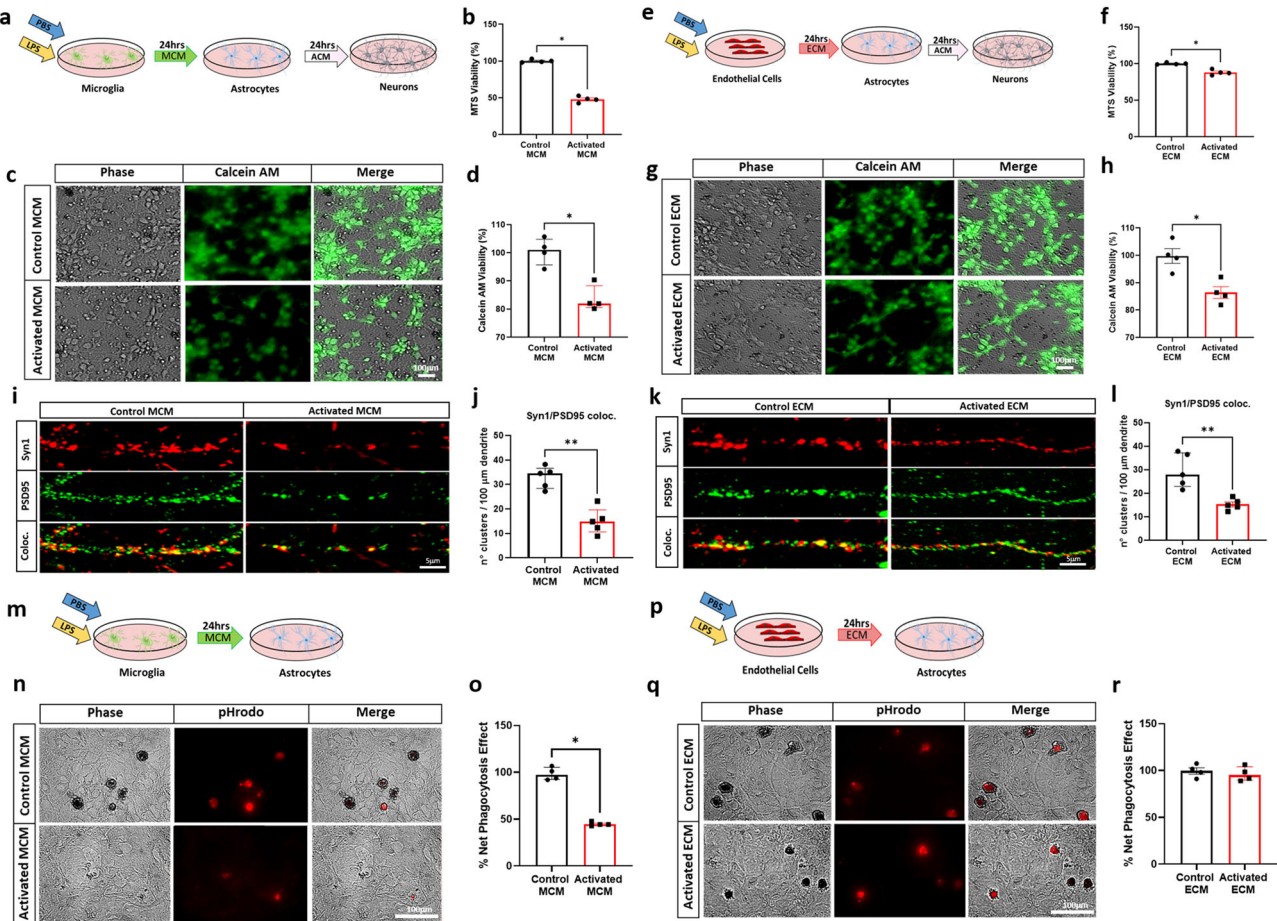

**Fig. 2 Astrocytes induced by activated ECM are neurotoxic and retain phagocytic capacity. a** Schematic diagram showing ACM treatment of neurons from previously treated control or activated MCM. **b** Quantification of MTS Viability assay **c**) Phase and fluorescence images of Calcein AM in cultured neurons (quantification in **d**). **e** Schematic diagram showing ACM treatment of neurons from previously treated control or activated ECM. **f** Quantification of MTS Viability assay **g** Phase and fluorescence images of Calcein AM in cultured neurons (quantification in **h**). **i** Representative immunofluorescence photographs of clusters of the synaptic proteins Synapsin-1 (red), PSD95 (green), and their colocalization (yellow) following ACM treatment of neurons previously treated with control or activated MCM. **j** Graphs showing the quantification of positive clusters for the colocalization of Synapsin-1 and PSD95 per 100 µm of dendrite. **k** Representative immunofluorescence photographs of clusters of the synaptic proteins Synapsin-1 (red), PSD95 (green), and their colocalization (yellow) in ACM treatment of neurons from previously treated control or activated ECM. **l** Graphs showing the quantification of positive clusters for colocalization of Synapsin-1 and PSD95 per 100 µm of dendrite. Images were collected from a single dendrite per neuron and the number of dendrites analyzed was 10 per culture from 5 independent cultures. **m** Schematic diagram showing treatment of astrocytes with control or activated MCM. **n** Phase and fluorescence images of cultured astrocytes engulfing pHrodo-zymogen bioparticles (quantification in **o**). **p** Schematic diagram showing treatment of astrocytes with control or activated ECM **q** Phase and fluorescence images of cultured astrocytes engulfing pHrodo-zymogen bioparticles (quantification in **r**). The results are shown as the median ± IQR of n = ≥4. Asterisks indicate significant differences, where *P < 0.05, **p < 0.01 by Mann-Whitney test. Scale bar 5 µm or 100 µm respectively.

revealed an enrichment in genes corresponding to plasma membrane, cell surface and immune system (Fig. 4c and Supplementary Data 2). We then utilized IPA core analysis to identify perturbed gene networks in activated astrocytes. Remarkably, the top gene network identified in astrocytes treated with activated ECM was associated with extracellular matrix remodeling involved in collagen and laminin formation, such as *LUM*, *DCN*, *MMP12*, and *COL6A3* (Fig. 4d, e). This extracellular matrix network was only identified in astrocytes treated with activated ECM and not in the ones treated with activated MCM (Supplementary Data 2).

It is well known that activated microglia induce A1 astrocytes by secreting a pro-inflammatory cocktail comprising IL-1α, TNF-α and C1q[6]. To determine if activated endothelial cells secrete a similar pro-inflammatory cocktail, we measured the levels of IL-1α and TNF-α in microglial and endothelial conditioned media.

As previously reported, activated microglia enhance the levels of secreted IL-1α and TNF-α, but these pro-inflammatory factors were not detected in endothelial conditioned media (Supplementary Fig. 4), suggesting that activated endothelial cells secrete other factors to induce reactive astrogliosis. Considering the extracellular matrix profile of endothelial-induced astrocytes, we evaluated the levels of MMP2, MMP12, GDF-15, and CD93, known proteins to be involved in the extracellular matrix signaling[20–22]. The level of these proteins was not increased in the media from activated endothelial cells versus that from control cells (Supplementary Fig. 4). Further studies will be necessary to determine the mediators secreted by endothelial cells that activate astrocytes into a neurotoxic phenotype.

Overall, these results suggest that, despite the shared induction of A1 astrocytic genes, the transcriptomes and mediators of endothelial-induced astrocytes are different from those activated

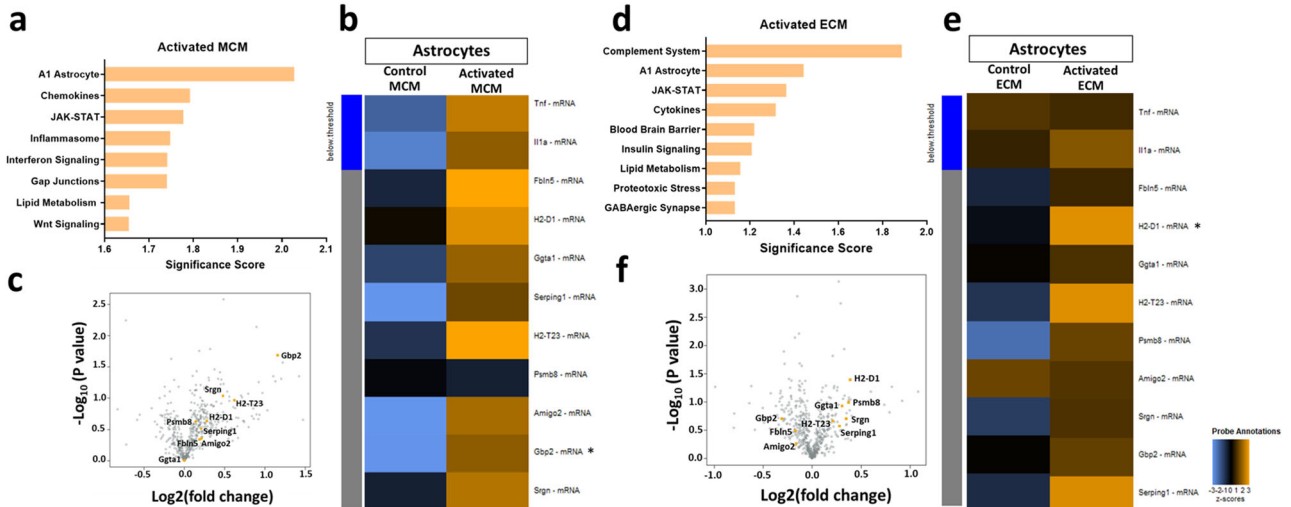

**Fig. 3 Endothelial activated astrocytes adopt an A1-like astrocytic profile. a** NanoString Gene Set enrichment analysis significance scores of astrocytes treated with activated MCM vs. control MCM **b** Unsupervised clustering of global undirected significance score of A1 astrocytic genes of activated MCM vs. control MCM treated astrocytes **c** Volcano plots showing directed differentially expressed genes in activated MCM vs. control MCM-treated astrocytes **d** NanoString Gene Set enrichment analysis significance scores of astrocytes treated with activated ECM vs. control ECM **e** Unsupervised clustering of global undirected significance score of A1 astrocytic genes in activated ECM vs. control ECM treated astrocytes. **f** Volcano plots showing directed differential expression of activated ECM vs. control ECM treated astrocytes. $n = 3$. In **b** and **e**, asterisks indicate significant differences, where *$P < 0.05$ by Mann-Whitney test. Due to the small sample size and exploratory purpose, statistical significance is reported by raw p values, and is not corrected by FDR.

by microglia. Interestingly, in the first study in which A1 astrocytes were identified in mice injected with LPS, a strong induction of several extracellular matrix genes, including *Dcn*, was observed[23], suggesting that endothelial-induced astrocytes are a subtype of reactive astrocytes distinct from those activated by microglia.

**Reactive endothelial cells induce Decorin⁺ C3⁺ astrocytes**. To identify if molecular signatures stratify astrocytes activated by microglia and endothelial cells, we performed immunostaining for C3, a putative A1 astrocyte marker, and GBP2, a highly upregulated marker in astrocytes activated by microglia[6]. We observed that astrocytes treated with activated MCM had a 52% increase in GBP2⁺ area in comparison to control treatments (Fig. 5a, b); however, no change in GBP2⁺ area was observed in astrocytes treated with activated ECM (Fig. 5c, d), supporting our NanoString and RNA-seq results (Fig. 4, GEO: GSE175485, GSE180106). Through RNA-seq analysis of unique genes upregulated in astrocytes treated with activated ECM, we identified Decorin (*Dcn*) as an extracellular matrix gene of particular interest, with known roles in extracellular matrix remodeling and as a ligand of various cytokines and growth factors, directly or indirectly interacting with signaling molecules and playing a vital role in proinflammatory processes[24]. Interestingly, *Dcn* is upregulated in A1 astrocytes isolated from LPS-injected mice[23]. We immunostained astrocytes for Decorin and C3 and observed no change in the Decorin⁺ area in astrocytes treated with activated MCM (Fig. 5e, f); however, astrocytes treated with activated ECM demonstrated a 27% increase in Decorin⁺ area in comparison to that of astrocytes that received control treatments (Fig. 5g, h). These observations demonstrate that LPS-activated endothelial cells are responsible for the induction of Decorin⁺ astrocytes and how Decorin and GBP2 stratify the molecular subtypes of C3⁺-astrocytes.

**Decorin⁺ C3⁺ astrocytes are present in a mouse model of cerebral amyloid angiopathy**. CAA is characterized by the

cerebrovascular deposition of amyloid[25]. Experimental evidence suggests that the nonmechanical consequences of CAA are responsible for cerebrovascular dysfunction and are mediated via endothelial cell pathways[26]. Therefore, to identify the presence of Decorin⁺ C3⁺ astrocytes in vivo, we performed immunohistochemical analyses in the Tg-FDD mouse model characterized by vascular amyloid accumulation. We recently demonstrated the presence of C3⁺-astrocytes associated with vascular amyloid deposits in the Tg-FDD model[11]. We also evaluated brain sections of APP/PS1 mice characterized by the preponderant accumulation of amyloid plaques and progressive vascular amyloid accumulation[27–29]. In this model, C3⁺-astrocytes have been identified to be associated with parenchymal amyloid deposits[30]. When we evaluated endothelial cell damage by ICAM-1 staining, we observed a significant increase in ICAM-1⁺ area associated with vascular pathology in Tg-FDD mice (Supplementary Fig. 5a, b). In APP/PS1 mice, although not significant, ICAM-1⁺ area was observed, in areas with vascular amyloid deposits (Supplementary Fig. 5a, b), suggesting that vascular amyloid accumulation could induce endothelial cell reactivity. To determine if Decorin⁺ astrocytes are associated with vascular amyloid accumulation in Tg-FDD and APP/PS1 mice, we performed triple staining with Thio-S, GFAP, and Decorin. Our results showed that 37% of the astrocytes associated with vascular amyloid in Tg-FDD mice were Decorin⁺. We also showed that Decorin⁺ astrocytes are not associated with parenchymal plaques in the APP/PS1 mice; however, there was an increase in Decorin⁺ astrocytes associated with vascular amyloid accumulation in APP/PS1 mice. Fifteen percent of the astrocytes associated with vascular amyloid in APP/PS1 mice were Decorin⁺. Decorin⁺ astrocytes were rarely observed in WT mice (Fig. 6a, b). To establish if these Decorin⁺ astrocytes were also C3⁺, we performed triple immunostaining with GFAP, C3, and Decorin antibodies. Quantitative analysis throughout the cortex revealed that in Tg-FDD 24% of astrocytes were C3⁺ and Decorin⁺ and 60% of astrocytes were C3⁺ and Decorin⁻. In APP/PS1 mice, 7% of the astrocytes were C3⁺ and Decorin⁺ and 46% of the astrocytes were positive for C3 but not Decorin. Astrocytes positive for C3 and Decorin were rarely

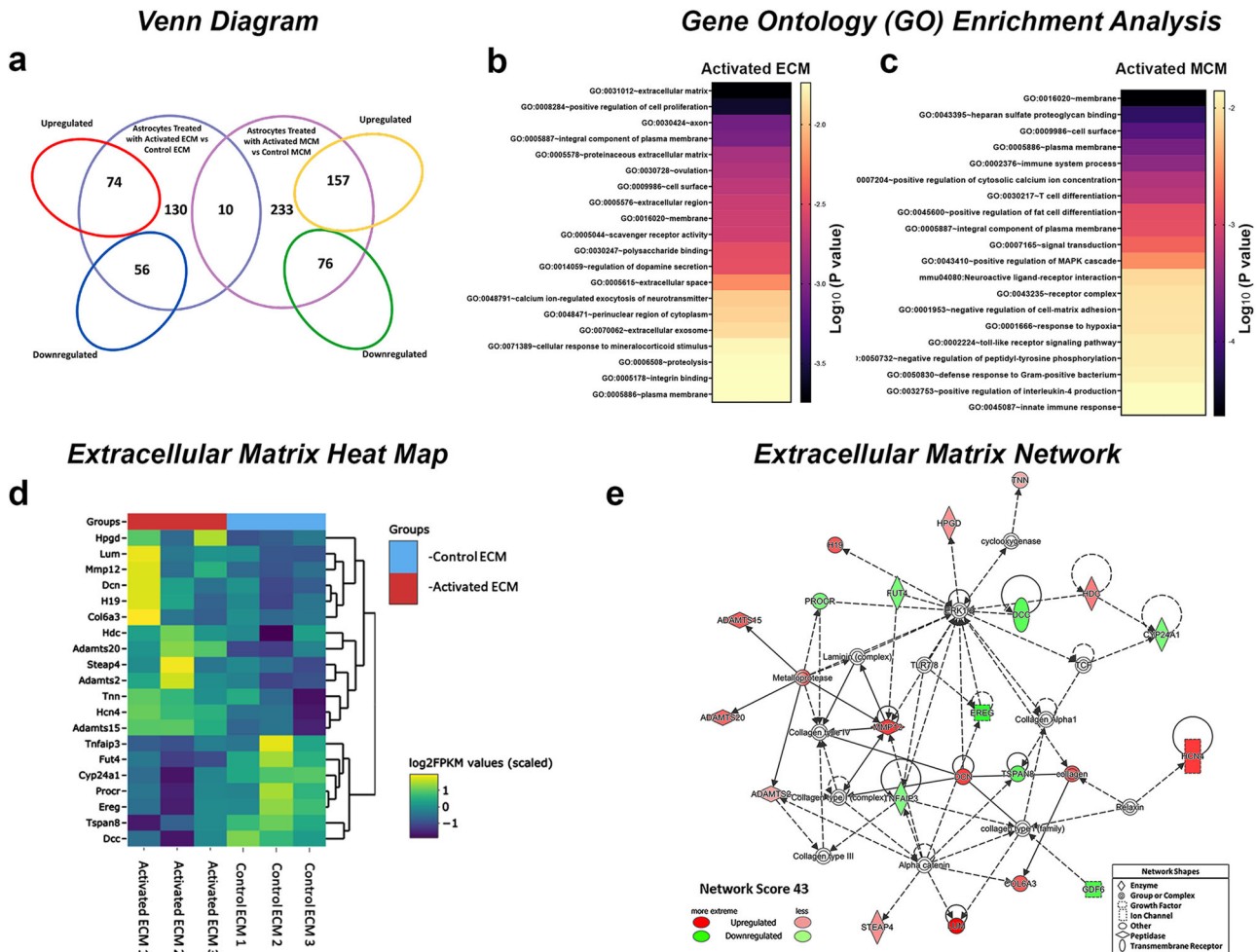

**Fig. 4 Endothelial activated astrocytes are enriched in extracellular matrix remodeling genes. a** Venn diagram of the RNA-seq analysis showing the intersection of genes enriched in transcriptomes of astrocytes treated with activated MCM and activated ECM. **b, c** DAVID bioinformatics gene ontology enrichment analysis of differentially expressed genes upregulated in astrocytes treated with activated ECM and activated MCM **d** RNA-seq generated heatmap of extracellular matrix gene expression (yellow = upregulation, purple = downregulation) and **e** network analysis of top scored network by ingenuity pathway analysis from astrocytes treated with activated ECM. $n = 3$. Due to the small sample size and exploratory purpose, statistical significance is reported by raw p values, and is not corrected by FDR.

observed in the WT controls (2%) (Fig. 6c, d). Interestingly, we did not identify astrocytes positive for Decorin and negative for C3 in any of the mice, suggesting that Decorin+ astrocytes could be a subtype of C3+astrocytes. The fact that Decorin+ astrocytes make up a large proportion of C3+ astrocytes in Tg-FDD mice, a model of vascular pathology, major neuroinflammation[11] and endothelial cell damage, suggests that these Decorin+ C3+ astrocytes could be associated with endothelial cell activation.

**Decorin-positive astrocytes in AD/CAA patients**. To evaluate if Decorin induction is conserved in human astrocytes, we performed triple immunofluorescence analysis of CAA, AD, and non-AD postmortem brain samples (Supplementary Table 1). Decorin+ astrocytes were rarely found in control tissue; however, we found that in all human AD/CAA patient brain samples analyzed, Decorin+ astrocytes were associated with vascular amyloid accumulation but not with parenchymal amyloid plaques (Fig. 7). Colocalization analysis showed that Decorin+ GFAP+ signals were only observed within astrocytes associated with vascular amyloid deposits. Both orthogonal and three-dimensional confocal views confirmed that Decorin+ GFAP+ signal was indeed surrounding vascular amyloid deposits. Decorin+ astrocytes were also confirmed to be C3+ in human AD/

CAA patient samples (Supplementary Fig. 6). Furthermore, GBP2+ astrocytes were observed only in the vicinity of parenchymal amyloid deposits but not near vascular amyloid deposits in AD/CAA patient samples, supporting the notion that reactive astrocytes associated to parenchymal amyloid deposits are distinct from astrocytes associated with vascular amyloid deposits (Supplementary Fig. 7). These findings suggest that cerebrovascular dysfunction due to vascular amyloid deposits, possibly mediated by endothelial cells, can induce reactive astrocytes distinct from astrocytes classically activated by proinflammatory microglia and demonstrates the existence of potentially extensive and subtle functional diversity among C3+ reactive astrocytes.

**Discussion**
In the present study, we showed that LPS-mediated inflammation of endothelial cells induces the formation of C3+-reactive astrocytes with a strong extracellular matrix remodeling profile. These C3+-reactive astrocytes are distinct from those classically activated by proinflammatory microglia. Furthermore, we showed that activated endothelial cells strongly induce C3+ astrocytes enriched in microglia-induced A1 astrocytic genes and exhibit a gain of neurotoxic function while retaining homeostatic

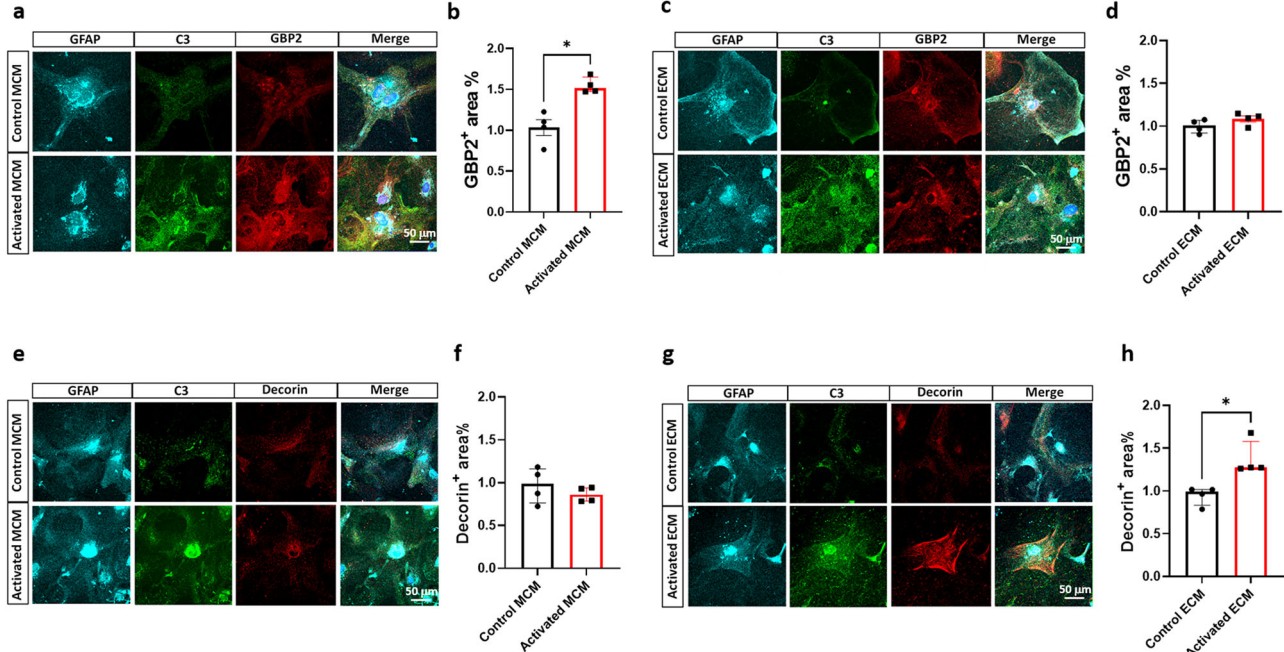

**Fig. 5 Reactive endothelial cells induce C3+ astrocytes distinct from reactive microglia. a, c** Triple immunofluorescence of astrocytes (GFAP, cyan), complement component 3 (C3, green) and GBP2 (red) in primary astrocyte cultures. **b, d** Quantification of GBP2+ area (%) of control or activated MCM/ECM-treated astrocytes, an increased presence of GBP2 immunoreactivity is observed only in astrocytes stimulated with activated MCM. **e, g** Triple immunofluorescence of astrocytes (GFAP, cyan), complement component 3 (C3, green) and Decorin (red) in primary astrocyte cultures. **f, h** Quantification of Decorin+ area (%) of control or activated MCM/ECM-treated astrocytes, an increased presence of Decorin immunoreactivity is observed exclusively in astrocytes stimulated with activated ECM. The number of cells analyzed was 20 per image from 10 different images per culture from 4 independent cultures. Results are shown as the median ± IQR of $n = 4$. Asterisks indicate significant differences were, *$P < 0.05$ by Mann-Whitney test. Scale bar 50 μm.

phagocytic capacity. Additionally, the transcriptomes of astrocytes differ when induced by endothelial and microglial inflammatory stimuli and stratify to molecular subtypes of C3+ reactive astrocytes. Finally, we demonstrated how astrocytes adopt a C3+ Decorin+ phenotype when exposed to LPS-treated endothelial cells. These C3+ Decorin+ astrocytes are associated in vivo with vascular amyloid deposits but not with parenchymal amyloid plaques in mouse models and AD/CAA patient samples. The exact mediators secreted by endothelial cells that induce C3+ Decorin+ astrocytes were not identified due to the small number of possible mediators analyzed. Further studies will be necessary to determine the exact factors responsible for the induction of C3+ Decorin+ astrocytes. These findings, suggest the existence of potentially extensive and subtle functional diversity among C3+-reactive astrocytes and the multidimensional roles of reactive astrocytes after CNS injuries and in neurodegenerative diseases.

Recently, transcriptomic profiling has helped identify the diverse heterogeneity and distinct molecular states of astrocytes in different disease models[18]. In an early transcriptomic study[23] and its follow-up[6], it was proposed that astrocytes adopt a neurotoxic phenotype characterized by increased expression of C3 after exposure to specific cytokines secreted by microglia exposed to lipopolysaccharide (LPS), whereas they acquire a neuroprotective phenotype in the middle cerebral arterial occlusion (MCAO) model of ischemic stroke. Recently, the heterogeneity of neuroinflammatory astrocyte subtypes in-vivo after LPS challenge has been clarified at single-cell resolution, indicating widespread responses and distinct inflammatory transitions of astrocytes with defined transcriptomic profiles in response to inflammation[7]. Furthermore, it has become increasingly clear that these binary A1 and A2 divisions are not fixed, and that neurotoxic and neuroprotective designations are not all-encompassing, as

blocking A1 astrocyte conversion is neuroprotective in ALS and Parkinson's disease models[9,31]. However, the abolishment of A1 astrocytes in a mouse model of prion disease led to an accelerated disease course with early dysregulation of microglia, which the authors suggested was due to the existence of a prion-induced specific subtype of C3+-reactive astrocytes[19]. Additionally, data from human patients with sporadic Creutzfeldt-Jakob Disease (sCJD) show that A1 astrocyte markers, such as GBP2, stratify to molecular subtypes in sCJD brains, which may account for the divergence of A1 astrocytic markers among molecular subtypes in these patients[32]. These findings indicate a much more complex disease-specific interplay than previously recognized.

Several recent studies have highlighted the importance of the functional architecture of the brain endothelial cells that form the BBB and astrocytes that support the function of neuronal populations and communicate with associated segments of the microvasculature, forming well-structured neurovascular units[33,34]. Damage to the neurovascular unit disrupts the endothelium and BBB integrity, promoting neuronal death, neuroglial dysfunction, and neuroinflammation[35]. Astrocytes play a primary role in forming the mature brain vasculature, maintaining BBB integrity, maintaining cerebral blood flow and secreting several proteins that remodel the extracellular matrix surrounding neurons and synapses that participates in numerous beneficial and deleterious roles[36,37]. In response to tissue damage, inflammation, or disease, the expression levels of various extracellular matrix molecules are highly upregulated in astrocytes, and major depositions are observed, often marking lesion sites[38]. The role of extracellular matrix remodeling by reactive astrocytes has been studied in detail, as astrocyte modifications of the extracellular matrix can limit the spread of damage after injury by the formation of glial scars but can also inhibit axon regeneration[38]. In

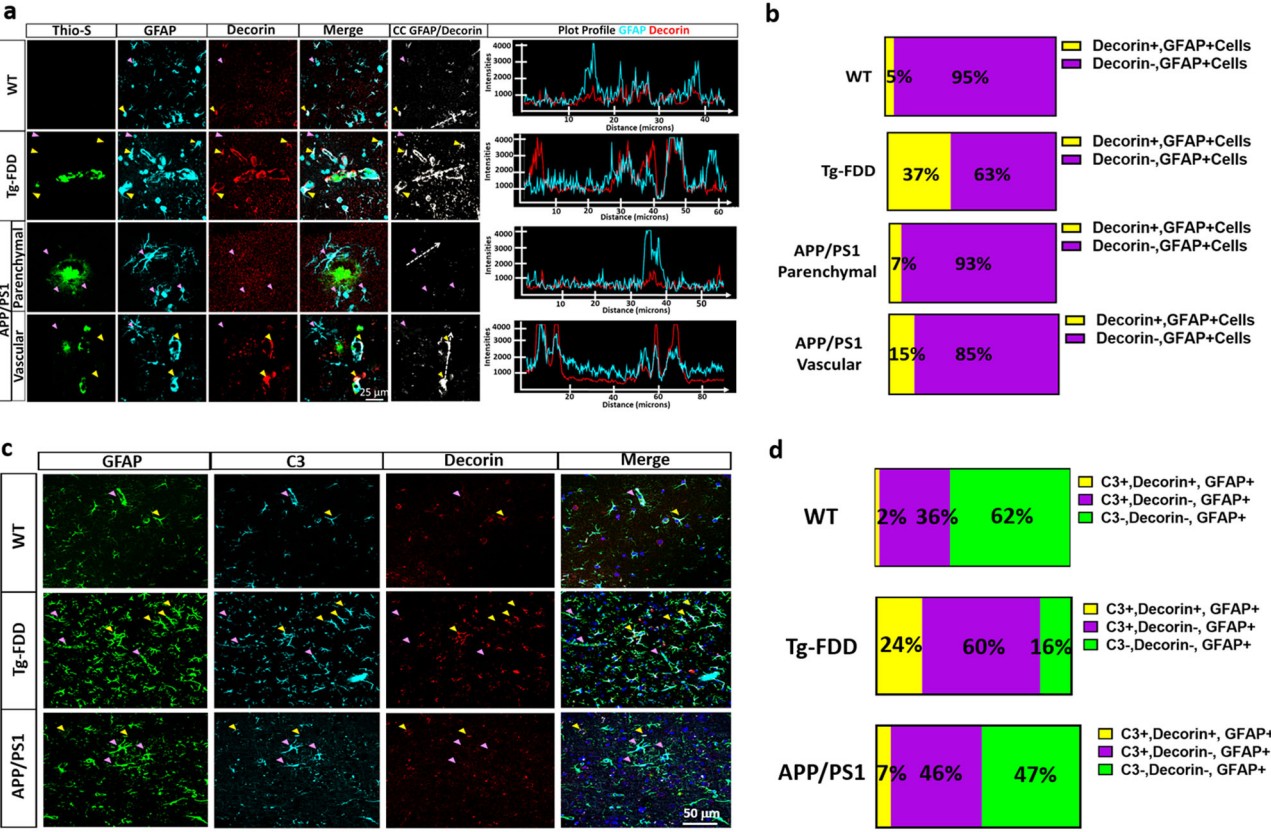

**Fig. 6 Decorin+ astrocytes are associated with vascular amyloid in vivo. a** Triple immunofluorescence of amyloid (Thio-S, green), astrocytes (GFAP, cyan), and Decorin (red) in the cortex of 15-month-old Tg-FDD, APP/PS1 mice and WT controls. Major presence of Decorin immunoreactivity is observed in the perivascular region of Thio-S positive vasculature in Tg-FDD mice. Yellow arrows indicate Decorin+, GFAP+ cells and purple arrows indicate Decorin-, GFAP+ cells. Colocalization analysis (CC) was performed to determine pixel intensity correlation between GFAP and Decorin. White pixels indicate colocalization between GFAP and Decorin signal and white dashed arrows represent distance in plot profiles showing overlapping intensities of astrocytes (GFAP, cyan) and Decorin (red). **b** Quantification of the proportion of Decorin+ cells in Tg-FDD, APP/PS1 and WT mice. **c** Triple immunofluorescence of astrocytes (GFAP, green), complement component 3 (C3, cyan), and Decorin (red) in the cortex of 15-month-old Tg-FDD and APP/PS1 mice. Yellow arrows indicate C3+, Decorin+, GFAP+ cells and purple arrows indicate C3+, Decorin-, GFAP+ cells. **d** Quantification of the proportion of C3+Decorin+ astrocytes versus C3+ astrocytes in Tg-FDD, APP/PS1 and WT mice. For quantifications, eight to ten images were used for each experiment ($n = 3$), and 200 cells were counted. All are representative images of 15-month-old Tg-FDD, APP/PS1 or WT mice. Scale bar 25 or 50 μm, respectively.

accordance with this preponderant effect of reactive astrocytes on the extracellular milieu, Gene Ontology grouping of astrocytes isolated after systemic LPS injection and MCAO revealed extracellular matrix binding and modifications as the top enriched networks[23]. Although LPS-reactive astrocytes and MCAO-reactive astrocytes each displayed an extensive induction of genes involved in modifying the extracellular space, the particular genes within each class differed[23]. According to the authors, this is likely due to differences in the inducing signal for each of these subtypes of reactive astrocytes[23]. For instance, an over 4-fold increase in *Dcn* was identified only in response to neuroinflammation after systemic LPS injections, suggesting Decorin as a marker for reactive neurotoxic astrocytes[23]. Decorin expression is also upregulated in astrocytes following spinal cord injury, a type of injury in which the primary insult disrupts the local vasculature, leading to immediate vascular disruptions at the injury epicenter, with most of the necrotic damage incurred by endothelial cells[39,40]. Although LPS itself mostly fails to cross the BBB, LPS injection causes extensive endothelial cell activation and neuroinflammation, causing transcriptional changes favoring the production of prostaglandins, proinflammatory cytokines, reactive oxygen species and nitrogen species, while leaving brain structures intact[41,42]. The reactive astrocytes identified in LPS-

injected mice[23] are potentially not solely astrocytes activated by microglia but also those activated by endothelial cells. This finding is supported by the results of our study, in which we demonstrated how LPS-induced inflammatory endothelial cells can induce C3+-reactive astrocytes with a prominent extracellular matrix remodeling profile and that these astrocytes are distinct from those classically activated by proinflammatory microglia, showing subtle functional diversity of neurotoxic reactive astrocytes and suggesting that the inflammatory stimuli of other cell types in the CNS may also induce a variety of C3+-reactive astrocytes, as originally proposed[43].

The aggregation and accumulation of vascular amyloid has long been recognized as a major contributor to neuroinflammation, playing a fundamental role in the pathogenesis of CAA and AD[25]. We and others have reported severe astrogliosis with a C3+ astrocytic phenotype without reactive microgliosis in CAA, suggesting that other cell types may be responsible for C3+ astrocytic induction associated with vascular amyloid deposits, independent of microglia activation[11-13]. Interestingly, in CAA, endothelial cells are well preserved. Even in severe cases, in which the mechanical consequences of CAA include intracerebral or subarachnoid hemorrhage, only a small fraction of CAA results in severe bleeding; however, the nonmechanical consequences of

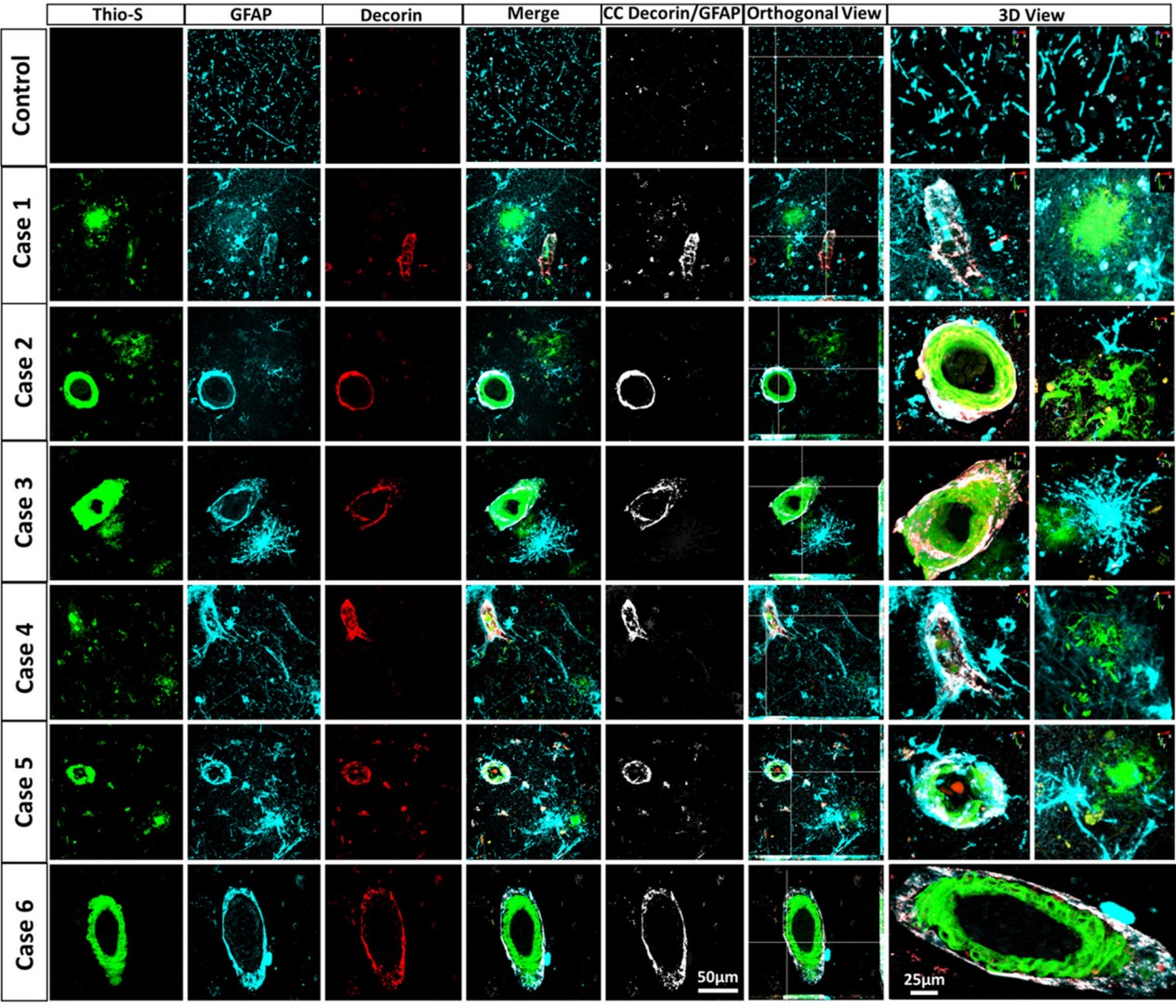

**Fig. 7 Decorin+ astrocytes are associated with vascular amyloid in AD/CAA patients.** Triple immunofluorescence for amyloid (Thio-S, green), astrocytes (GFAP, Cyan) and Decorin (red) in AD/CAA and healthy control patients show Decorin+ staining associated with vascular amyloid but not parenchymal amyloid deposits. Thio-S, GFAP and Decorin immunoreactivity overlay (Merge). Colocalization analysis (CC) was performed to determine pixel intensity correlation between Decorin and GFAP. White pixels indicate colocalization between Decorin and GFAP signal. The right panel shows orthogonal images and reconstructed three-dimensional views. Scale bar 25 or 50 μm, respectively.

CAA in cerebrovascular regulation are prevalent, and there is strong evidence that endothelial-dependent factors that disrupt cerebrovascular regulation are even more deleterious[26]. Recent studies have demonstrated the effect of Aβ amyloid on endothelial cell dysfunction, indicating that Aβ fibrillar deposits are responsible for inhibiting angiogenesis and increasing BBB permeability without inducing apoptosis, linking the biological effect of Aβ aggregation with the clinical aspects of CAA and endothelial dysfunction[44,45]. Furthermore, cerebral endothelial cells remain ultrastructurally intact in APP mice and human CAA patient samples, and studies using isolated cerebral arteries have indicated that brain parenchyma participation is not required for Aβ-associated vascular dysregulation, which suggests a direct effect of amyloid on vessels[46–48]. It has been suggested that changes in common pathways utilized by endothelium-dependent vasoactive factors, rather than endothelium structural problems, cause this endothelial dysfunction in CAA. Therefore, considering these findings regarding endothelial cell dysfunction associated with CAA and our primary culture-based results that show that endothelial cell activation and dysfunction is responsible for

inducing C3+Decorin+ astrocytes, it is reasonable to suggest that the Decorin+ astrocytes associated with vascular amyloid deposits in mouse models and AD/CAA patients could be induced by activated endothelial cells. The presence of Decorin+ astrocytes associated with vascular amyloid deposits strongly suggests that vascular damage induces C3+ astrocytes distinct from astrocytes classically activated by proinflammatory microglia and demonstrates the existence of potentially extensive and subtle functional diversity among C3+-reactive astrocytes.

Overall, our findings suggest that distinct subtypes of C3+ reactive astrocytes exist depending on the activating cell type. We confirmed that C3+ GBP2+astrocytes are reactive astrocytes induced by microglia and identified C3+ Decorin+ astrocytes to be reactive astrocytes induced by endothelial cells. We further confirmed the in vivo presence of these C3+ Decorin+ astrocytes and that these cells are associated with vascular amyloid but not with parenchymal amyloid deposits in mouse models and AD/CAA patients. Finally, we provide evidence to suggest that these two subtypes of C3+ astrocytes are functionally different. Nevertheless, considering transcriptional changes were

determined using astrocytic cultures grown in media containing serum known to impact changes in gene expression and protein level, further transcriptomics studies using astrocytes grown in serum-free media which more closely mimic their acutely purified state[49] and brain samples will be necessary to dissect these differences in detail and, most importantly, to establish how many subtypes of C3+ astrocytes are present in the human brain under proinflammatory conditions, such as in neurodegenerative diseases.

## Methods

**Transgenic mouse model**. Tg-FDD, APP/PS1 (JAX stock #005864) and wild type C57/BL6J (WT) (JAX stock #000664) male and female mice were used in our experiments, including for cellular and immunohistochemistry (IHC) analyses. The Tg-FDD mouse model expresses an FDD-associated human mutant BRI2 transgene that leads to the vascular accumulation of ADan amyloid[50]. The APP/PS1 mice are double-transgenic mice expressing a chimeric mouse/human amyloid precursor protein (Mo/HuAPP695swe) and a mutant human presenilin 1 (PS1-dE9), both directed to CNS neurons. The APP/PS1 model is characterized by the preponderant accumulation of parenchymal amyloid plaques and progressive vascular amyloid accumulation.[27–29] All mice were housed at the Indiana University School of Medicine (IUSM) animal care facility and were maintained according to USDA standards (12-h light/dark cycle, food and water *ad libitum*), per the Guide for the Care and Use of Laboratory Animals (National Institutes of Health, Bethesda, MD). Animals were anesthetized and euthanized according to IUSM Institutional Animal Care and Use Committee-approved procedures. Mice were deeply anesthetized prior to decapitation. For all described experiments, 15-month-old animals were utilized. This age was selected due to the extensive vascular amyloid accumulation at 15 months of age in the Tg-FDD model as well as parenchymal and vascular amyloid deposition in the APP/PS1 model[11,27–29,51,52].

**Brain sections immunofluorescence**. Paraffin sections were deparaffinized in xylene, rehydrated in ethanol (EtOH) and washed with deionized water. Then, the sections were heated in a microwave oven in low pH antigen retrieval solution (eBioscience) for 4 min twice. After washing in PBS for 5 min twice, the sections were blocked with PBS 5% goat serum, 5% horse serum, 2% fish gel, and 0.01% Triton X-100 for 1 h at room temperature (RT). Sections were then incubated overnight at 4 °C with the following antibodies: anti-GFAP (G3893, Sigma-Aldrich), anti-IBA1 (1019-19741, Wako), anti-CD 31 (PECAM-1) (1:100, PA5-16301, ThermoFisher), anti-ICAM-1 (MA5407, ThermoFisher) anti-C3 (PA5-21349, ThermoFisher), anti-Decorin (14667-1-AP, ThermoFisher), and anti-GBP2 (LS-C114752) diluted 1:100 in blocking solution. The next day, the sections were quickly washed 3 times in PBS and incubated with 1:500 biotinylated horse anti-mouse antibody (BA-200, Vector) and/or biotinylated goat anti-rabbit antibody (BA-1000, Vector) for 1 h at RT. Thirty minutes in advance, the Vectastain Elite ABC peroxidase kit (PK-6100, Vector) was prepared according to the manufacturer's instructions. After the secondary antibody incubation, the sections were incubated with the A + B solution for 30 min at RT. After 3 quick washes with PBS, tyramide dyes were prepared at 1:500 in PBS, and the slides were incubated with the dyes for 10 min at RT. Next, slides were incubated with 3% $H_2O_2$ for 10 min at RT to stop peroxidase activity. Amyloid was stained with 1% Thioflavin-S for 8 min at RT, followed by two 3-min washes in EtOH 50 and 30% and a final 5-min wash in deionized water. Finally, the sections were washed in PBS and mounted with Vectashield mounting medium with or without DAPI (Vector Laboratories).

**Neuron, astrocyte, microglia, and brain microvascular endothelial primary cell culture**. The procedure used for cortical neuronal cultures was based on a previously described approach[53]. Briefly, the brains were extracted and the cortexes were dissected and washed twice with dissection medium (DM) (97.5% HBSS, 1X sodium pyruvate, 0.1% glucose, 10 mM HEPES). The cortexes were suspended in 4.5 mL DM and incubated with 2.5% trypsin and 1% DNAse for 15 min. After two washes with DM, cortexes were washed twice with 37 °C prewarmed plating medium (PM) (86.55% MEM Eagle's with Earle's BSS, 10% filtered and heat-inactivated FBS, 0.45% glucose, 1X sodium pyruvate, 1X glutamine, and 1X penicillin/streptomycin). Tissue was desegregated with glass Pasteur pipettes with tips previously rounded by direct flaming. After desegregation, a cell strainer (40 μm pore) was used. The cells were counted, and 200,000 cells/mL were seeded in 12 well plates containing 1 mL of 37 °C prewarmed maintenance medium (MM) (95% neurobasal medium, 1x B-27 supplement, 1x glutamine, and 1x penicillin/streptomycin) and 18 mm coverslips treated overnight with 0.5% poly-L-lysine in borate buffer. Five hours after seeding, half of the medium was replaced with fresh MM prewarmed at 37 °C. On the 3rd day of culture, cytosine arabinoside (AraC) was added at a final concentration of 3 μM. Neurons were maintained for 14 days, replacing half of the medium every two days with fresh 37 °C prewarmed MM. Primary glial cell cultures were prepared from the brain cortexes of newborn (postnatal day 0–3) C57/BL6J mice. Briefly, animals were euthanized, and their brains were extracted. The brains were cut into small pieces, collected in HBSS

(H9269, Sigma), and treated with 2.5% trypsin (15090-046, Gibco) and 1% DNase (EN0521, Thermo) at 37 °C for 15 min. Then, the tissue was disaggregated by pipetting, passed through a 70-μm pore cell strainer (352350, Corning), and collected in heat-inactivated FBS. Cells were centrifuged for 5 min at 1000× *g*, and the obtained pellet was resuspended in 10 mL glial medium (Advanced DMEM/F12 with 10% heat-inactivated FBS, 100 g/mL streptomycin, 100 UI/mL penicillin, and 200 μg/mL glutamine). Cells were then counted using a Luna Dual Fluorescence Cell Counter (Logos Biosystem). Cells were plated in 75-cm² cell culture flasks at a density of $1 \times 10^6$ and incubated until 90% confluent; the medium was changed every 2 days. For primary microglia; cell culture, confluent flasks were shaken for 1 h at 200 rpm at 37 °C to detach the microglial cells. Then, the supernatants containing microglia were collected and centrifuged for 5 min at 300× *g*. The cells were resuspended in 1 mL of glial medium, counted using an automatic cell counter (Logos Biosystem) and seeded in 6-well culture plates for further experiments. The flask was treated with 0.5% Trypsin (15400054, Gibco) for 10 min at 37 °C to detach the astrocytes. Then, the supernatant was collected, centrifuged at 300× *g* for 5 min, and the pellet was resuspended in glial medium. Astrocytes were quantified using an automated cell counter (Logos Biosystem) and seeded in 12-well ($2 \times 10^5$ cells) plates for further experiments. Primary mouse brain microvascular endothelial cells (BMECs) were isolated and cultured as previously described[54]. Briefly, after mouse brains were isolated from 8–10-week-old animals, the meninges-free forebrains were digested using 10 mg/mL collagenase CLS2 (17101015, Thermo) in DMEM for 60 min, then centrifuged at 4 °C for 20 min at 180 rpm. The pellets were resuspended in 1 mg/mL collagenase/dispase (Thermo) and incubated at 37 °C for 60 min on an orbital shaker. After the final wash, the resulting cells were cultured in endothelial cell medium. After 12 DIV cells were transitioned and maintained in serum-free media. Primary microglia and endothelial cells were treated with LPS (100 ng/ml) for 24 h. The next day, the medium was replaced with fresh serum-free medium and conditioned for another 24 h, defined as microglia conditioned medium (MCM) and endothelial conditioned medium (ECM). Control or activated conditioned media were then collected and concentrated using a Speed Vac Plus SC110A and resuspended in 500 μL of PBS. The total protein concentration was determined using the Pierce BCA protein assay kit (23246, Thermo Scientific), and 50 μg of total protein was added to mouse primary astrocytes. The endotoxin content of the conditioned media was quantified using Pierce Chromogenic Endotoxin Quantification kit (A39552, Thermo) following the manufacturer's instructions. The amount of endotoxin in conditioned media was under one endotoxin unit (EU)/ml, a concentration that is incapable of inducing significant glial activation as previously reported[55–58]. For every culture, cellular shape, confluency and overall health was evaluated on a daily basis. Cultures that did not meet the criteria of our internal controls were discarded immediately and were not used.

**Cell culture immunofluorescence and purity assessment**. Neurons, astrocytes, microglia or brain microvascular endothelial cells were seeded on 18 mm diameter coverslips. Once the culture reached 90% confluence, the cells were fixed in PBS containing 4% paraformaldehyde (PFA) for 15 min. Cells were permeabilized for 5 min at RT in 0.25% Triton X-100 in PBS, washed twice with PBS, and incubated for 1 h at 37 °C in PBS containing 5% goat, 5% horse serum and 2% fish gel (blocking solution). Cells were then incubated overnight at 4 °C in primary antibodies diluted 1:100 anti-GFAP (G3893, Sigma-Aldrich), anti-IBA (1019-19741, Wako), anti-C3 (PA5-21349, ThermoFisher), anti-Synapsin-1(ab64581, Abcam), anti-PSD95 (ab2723, Abcam), anti-Decorin (14667-1-AP, ThermoFisher), anti-GBP2 (PA5-112426, Thermo) and anti-TREM2 (AF1729, R&D) in blocking solution. After incubation, cells were washed with PBS, then incubated with Alexa 488-conjugated goat anti-mouse antibody (1:100, Invitrogen), Alexa 568-conjugated goat anti-rabbit antibody (1:100, Invitrogen) or Alexa 647-conjugated goat anti-mouse antibody (1:100, Invitrogen) for 1 h at RT, washed with PBS and mounted with Vectashield mounting medium with or without DAPI (Vector Laboratories). To assess the purity of astrocytes, microglia or endothelial culture, the percentage of GFAP-positive, IBA1-positive or Pecam-1 positive cells in the total cells was determined. Samples were examined using a Nikon A1-R laser scanning confocal microscope coupled with Nikon AR software. At least 20 cells were analyzed from each image. Ten images were used for each experiment, and four to five independent culture experiments were performed.

**Microscopy and image analysis**. For cell cultures and mouse brain sections, we used ImageJ software (NIH) to create one index that represented changes in the number of PECAM-1 +, ICAM-1+, Decorin+, GBP2+, TREM2+, C3+, GFAP+ or IBA1+ pixels divided by the total number of pixels in the image, expressed as (+) area %[59]. Tg-FDD and APP/PS1 (4 animals per genotype) cerebral cortexes were examined using a Leica DMi 8 epifluorescence microscope coupled with the LAS X program (Leica) or Nikon A1-R laser scanning confocal microscope coupled with Nikon AR software (Nikon).

**Phagocytosis assay**. Astrocytes were grown for 7 days then treated with 50 μg total protein from control or activated MCM and ECM for 24 h. A phagocytosis assay was performed as per the supplier's instructions with pHrodo™ Red Zymosan A BioParticles® (Life Technologies, Carlsbad, USA) conjugate for phagocytosis. In

brief, astrocytes were plated on a 96-well tissue culture plate and incubated in MEM containing fluorescently labeled zymosan particles (0.5 mg/ml) for 4 h at 37 °C, 5 % $CO_2$. Fluorescence was measured with an Epoch2 microplate reader (BioTek).

*Survival/cell toxicity assay.* Astrocytes were grown for 7 days then treated with 50 µg total protein from control or activated MCM and ECM for 24 h. The next day, the medium was replaced with fresh serum-free medium and conditioning was conducted for another 24 h, defined as astrocytic conditioned medium (ACM). ACM was then collected and concentrated using a Speed Vac Plus SC110A and resuspended in 500 µL of PBS. The total protein concentration was determined using the Pierce BCA protein assay kit (23246, Thermo Scientific), and 15 µg of total protein was added to primary mouse cortical neurons (A15585, Thermo) cultured 5 DIV (10,000 cells per well in poly-d-lysine-coated 96-well plates) in neurobasal medium (21103, Thermo) supplemented with B27 (17504, Thermo) and GlutaMax$^{TM}$ (35050, Thermo). The cell viability was assessed after 24 h using the Live/Dead Kit for mammalian cells (Thermo Fisher Scientific, L3224) and the MTS cell titer 96 Aqueous One solution assay (G3582, Promega); absorbance and fluorescence were measured with FlexStation 3 (Molecular Devices) and the Epoch2 microplate reader (BioTek) respectively.

**NanoString gene expression analysis.** Total mRNA was purified from primary astrocytes and multiplexed using the nCounter analysis system (NanoString Technologies, Seattle, WA, USA) combined with the nCounter Mouse Glial Profiling Panel that includes 770 genes covering the core pathways that define glial cell homeostasis and activation. Briefly, 100 ng total RNA per sample (20 ng/µl) ($n = 3$) was loaded and hybridized with probes for 16 h at 65 °C following the manufacturer's protocol. All samples analyzed had a RIN value between 9.9 and 10. Counts for target genes were normalized to the best fitting house-keeping genes as determined by nSolver software to account for variation in RNA content. The background signal was calculated as the mean value of the negative hybridization control probes. The expression data were excluded when they had lower than average background signals from the negative controls, and probes with <100 reads for 6 or more samples were removed from the analysis. Significant genes for undirected differential expression were identified using Shapiro-Wilk test for normality followed by an unpaired t-test and identified by an * in the heatmap. Downstream analyses and visualizations of gene expression datasets were performed using ROSALIND analysis platform (OnRamp Bioinformatics) software.

**RNA sequencing and library preparation.** The quantity and quality of the total RNA were evaluated using Agilent Bioanalyzer 2100. All the samples ($n = 3$) had good quality with a RIN (RNA Integrity Number) of 9.9–10. One hundred nanograms of total RNA was used for library preparation. Briefly, cDNA library preparation included mRNA purification, RNA fragmentation, cDNA synthesis, ligation of index adaptors, and library amplification, following the KAPA mRNA Hyper Prep Kit Technical Data Sheet, KR1352–v4.17 (Roche Corporate). Each resulting indexed library was quantified, and the quality was accessed with a Qubit and Agilent Bioanalyzer, and multiple libraries were pooled in equal molarity. The pooled libraries were sequenced with 2×150 bp paired-end configuration on an Illumina NovaSeq 6000 sequencer using the v1.5 Reagent Kit.

**RNA-seq data analysis.** The sequencing reads were first assessed using FastQC (v.0.11.5, Babraham Bioinformatics, Cambridge, UK) for quality control. The sequence data were then mapped to the mouse reference genome mm10 using the RNA-seq aligner STAR (v.2.5)[60] with the following parameter: "—outSAMmapqUnique 60". To evaluate quality of the RNA-seq data, the read distribution across the genome was assessed using bamutils (from ngsutils v.0.5.9)[61]. Uniquely mapped reads were used to quantify the gene-level expression employing featureCounts (subread v.1.5.1)[62] with the following parameters: "-s 2 -Q 10". Genes with read count per million (CPM) < 0.5 in more than three samples were removed. The data were normalized using the TMM (trimmed mean of M values) method. Differential expression (DE) analysis was performed using edgeR (v.3.12.1)[63,64]. Raw $p$ value cutoff 0.05 was used as guidance to filter down the number of promising genes to a manageable size for further experimental validation.

**Genomic pathway analysis.** Ingenuity Pathway Analysis (IPA) core analysis was used to identify the perturbed gene networks in astrocytes treated with control or activated ECM/MCM. In the network figure generated by IPA, red represents upregulated genes and green represents downregulated genes; the main biological function of the network is provided. Then, a heatmap showing the differential expression of genes was generated for a specific perturbed gene network of interest to visualize the expression of the genes in the network. The DE cutoff for unique networks used for enrichment analysis was |foldchange| > 1.5 and raw $p$ < 0.05. GO and KEGG enrichment analyses were performed using the Database for Annotation, Visualization and Integrated Discovery (DAVID) website[65,66].

**Multiplex analysis.** The Mouse Magnetic Luminex Assay (EMD Millipore Corporation, Burlington, MA) was used for protein multiplex ELISA utilizing the Luminex xMAP technology (Luminex, Northbrook, IL) according to the manufacturer's protocol. The panel detects the proteins C1qR1/CD93, GDF-15, IL-1α, MMP-2, MMP-12 and TNF-α. Samples were thawed at room temperature and then spun at 10,000 g for 10 min. All samples ($n = 3$) were assayed in duplicate using a Bio-Plex™ 200 System with High Throughput Fluidics (HTF) Multiplex Array System (Bio-Rad® Laboratories, Hercules, CA) and analyzed using the Bio-Plex 6.0 Manager software (Bio-Rad).

**Alzheimer's disease and cerebral amyloid angiopathy patient samples.** Post-mortem brain tissues from patients with AD, CAA and healthy age-matched controls were provided in the form of paraffin sections by the Brain Resource Center at Johns Hopkins. Immunofluorescent analysis was performed on 15 µm paraffin sections from 7 patients with AD, CAA or combined pathology and healthy controls. The sections were examined using a Nikon A1-R laser scanning confocal microscope coupled with Nikon AR software (Nikon) for orthogonal images and reconstructed three-dimensional views.

**Statistics and reproducibility.** The details about experimental design and statistics used in different data analyses performed in this study are given in the respective sections of results and methods. Sample sizes were determined based on previous publications, and independent biological replicates range from 3 to 5 for all experimental modalities used in this study. The experimental analyses and data collection protocols were performed blind unless otherwise stated. Most data were analyzed by nonparametric measures and statistical comparisons were performed using a two-tailed Mann-Whitney test via GraphPad Prism as described. Data are presented as the median ± IQR unless otherwise stated. *, ** and *** denote $p$ < 0.05, $p$ < 0.01, and $p$ < 0.001, respectively.

**Reporting summary.** Further information on research design is available in the Nature Research Reporting Summary linked to this article.

## Data availability
The RNA-seq and NanoString data supporting the conclusions of this article are available at NCBI's Gene Expression Omnibus (GEO) and is accessible via series accession numbers GSE175485 and GSE180106, respectively. Source data behind the graphs in the main manuscript can be found in the Supplementary Data 3.

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

## Acknowledgements

We thank Dr. Louise Pay for her critical editing of the manuscript. We thank Abigail Perkins for her input in the study design, preparation, and analysis. RNA-seq studies were carried out by the Center for Medical Genomics at Indiana University School of Medicine, which is partially supported by the Indiana Genomic Initiative at Indiana University (INGEN); INGEN is supported in part by the Lilly Endowment, Inc. We thank the Multiplex Analysis Core at the Indiana University Melvin and Bren Simon Cancer Center for providing support in analyzing samples and interpretation of data. This work was supported by an NIH/NIA 1R01AG059639, 3R01AG059639, and AARGD-591887.

## Author contributions

C.A.L.-R. and X.T. conceived and coordinated the study; X.T. performed tissue collection, cell culture experiments, histological analyses, and biochemical analyses; P.C. performed synaptotoxicity experiments. N.J. and P.M. performed the MTS viability assay. Y.Y. assisted with NanoString and RNA-Seq. J.Z. and X.H. performed and coordinated the gene expression analysis from RNA-seq available datasets. J.R.-O. and J.T. provided human cases for study. R.V. provided animals for study; C.A.L.-R., X.T., S.X. and J.Z. analyzed data and drafted the images for publication; C.A.L.-R. and X.T. wrote the manuscript. All authors read and approved the final manuscript.

## Competing interests

The authors declare no competing interests.

## Ethics approval and consent to participate

All experiments conformed to the National Academy of Sciences Guide for Care and Use of Laboratory Animals and were approved by the Indiana University School of Medicine (IUSM) Animal Care and Use Committee.
