## [Peer Review File · Communications Biology]

Reviewers' comments:

Reviewer #1 (Remarks to the Author):

Activated Endothelial Cells Induce a Distinct 1 Type of Astrocytic Reactivity

In this manuscript, Taylor et al. describe an astrocyte reactivity state induced by activated endothelial cells that resembles a subtype previously described to be induced by activated microglia with some key differences. They show that while similar to microglia-induced astrocyte reactivity, activated endothelial conditioned medium (ECM) is neurotoxic and impairs astrocyte phagocytosis, the MCM and ECM induced astrocytes show differences in gene expression and protein levels. Finally, they show evidence that this ECM-induced subtype is associated with the inflamed vasculature in mouse models of Alzheimer's disease (AD) and in human AD tissue. Overall the findings are exciting and add an additional piece to the puzzle of (reactive) astrocyte heterogeneity.

However, I wonder whether rather than describing astrocytes as being of type "A1" it might be favorable to refer to them as microglia-induced? It is becoming increasingly clear that there are many more complex substates of reactivity that do not necessarily align with a simple A1/A2 dichotomy. In fact, it looks like based on Fig3 that maybe the authors are dealing with an Interferon-induced astrocyte subtype that has been described recently and is associated with the vasculature as well (PMID 34413515)? It might be worth comparing gene expression profiles of this Interferon-subtype and both MCM- and ECM-induced astrocytes described here? Do endothelial cells release Interferon upon LPS activation?

Experimentally, my main concern involves the RNA-seq analysis in Fig4. As far as I can tell, rather than using adjusted p-values, the authors used "raw" p-values and many genes highlighted (including all genes in Fig4f) have adjusted p-value cut-offs much higher than applied usually. Ideally, cutoffs should be 0.01-0.05. An FDR of 1 would indicate that 100% of the genes could be false-positives and should therefore not be included in downstream analyses.

Another concern is the culture system employed. The inclusion of serum, particularly as high as 10% in astrocyte culture media will irreversibly change the transcriptome and function of these cells. The authors must at least acknowledge this as a caveat of the approach. There is also no purity information provided for these cultures (of any cell type: astrocytes, microglia, or endothelial cells) - as the authors do not complete an endotoxin cleanup following their LPS treatments, any residual LPS carried over in conditioned media experiments could activate contaminating microglia in their astrocyte or endothelial cell cultures - leading to difficult to interpret effects. Given that endothelial TLR4 expression is considerably less (or not present?) when compared to microglia, this purity information becomes integral for all downstream analysis and interpretation. The endothelial response to peripheral inflammation driving astrocyte responses is an exciting and novel mechanism, but a lack of purity data and endotoxin cleanup in media exchange experiments remain a lynchpin of the entire model, I'd think this is required.

minor points:

Fig3: Maybe adding padj-values/FDRs or * to the heatmaps in b and e would make it clearer which

genes are significantly different across the two stimulations?

Fig4: Is it possible to add a heatmap of all ECM and MCM DEGs to really appreciate the different gene expression profiles?

Fig6: Did the authors perform a C3 staining of WT tissue? If so it would be worth including here.

Fig7: Similarly, is there a Decorin staining for non-AD tissue that could be added to this figure?

Reviewer #2 (Remarks to the Author):

This well written and carefully designed manuscript by Taylor et al. reports that injured endothelial cells can induce a unique A1 phenotype in astrocytes, which is characterized by a genetic signature associated with extracellular matrix remodeling factors. A1 astrocytes induced by activated endothelial cells exhibit elevated decorin and are associated with vascular A β deposits in animal and human models of CAA and AD, suggesting a subpopulation of reactive astrocytes, specifically associated with A β -induced endothelial cell dysfunction.

This work opens new avenues towards the possibility that multiple different subtypes of C3 positive astrocytes, likely activated by specific cell types, exist. This will allow to further comprehend if (and which) cell-specific molecular players may induce a characteristic pathological phenotype in astrocytes, which may mediate unique types of inflammatory responses in neurodegenerative diseases. Therefore, these findings are novel and informative, and appear to meet the scientific quality level to be suitable for publication in Communications Biology.

A few experimental revisions and text edits are suggested below:

1. It would be important to identify if major differences in cytokines within ECM and MCM may be responsible for the different activation of the astrocytes.
2. In figures 1-2-5 legends, please add how many cells and images were counted in each experiment. Most graphs represent 3 points. Please clarify what each point stands for.
3. Phase contrast images in Fig. 2, panels C, G, N, Q, at the lower magnification are not clear, please replace with higher quality images.
4. In Figure 2p, the word "astrocytes" is missing below the petri dish illustration.
5. line 124. The antibody is anti-IBA1.
6. Lines 551-556. Please divide or clarify sentence, which is a bit too long and therefore confusing.
7. line 858. In the legends of Figures 6a and 6c, please specify what the pink and yellow arrowheads point to. Moreover, in the legend of Figures 6a, explain the meaning of the white dashed arrows, in the CC GFAP/decorin images.
8. Figure 7. In human brains, the authors may consider to co-stain also for C3, GFAP and decorin, to

confirm the A1 astrocytic phenotype in decorin positive cells. Additionally, co-staining with C3, GBP2 and decorin would be useful to confirm that these A1 astrocytes are GBP2 negative.

9. Figure 7. Please add scale bar in 3D view.

Reviewer #3 (Remarks to the Author):

The work by Taylor and colleagues investigates whether activated endothelial cells can induce neurotoxic reactive astrocytes. Authors demonstrated that activated endothelial cells induce upregulation of C3 in astrocytes, a hallmark of A1 astrocytes induced by microglial activation. However, they show that C3+ neurotoxic astrocytes induced by activated endothelial cells present a distinct profile than astrocytes activated by microglia. Finally, they demonstrate that endothelial-activated astrocytes are Decorin-positive and seem to be associated with vascular amyloid deposits in mouse models and AD/CAA patients. These findings build upon the idea of astrocyte reactivity heterogeneity and are of interest. The article is well written, and experiments seem of high technical quality. However, some points should be addressed before a potential publication in *Communications Biology*.

1. The use of A1 to characterize a universal neurotoxic astrocyte phenotype has been widely debated. The recent consensus article proposed to avoid using it (Ref 1 – Escartin et al.). I would advise authors to make a more balanced introduction and discussion about astrocyte heterogeneity.
2. Statistics used for evaluating RNA-seq data are poorly described. For example, differentially expressed genes are defined as (Fold change > 1.5, $p < 0.05$). Are these data unadjusted? They mention FDR correction in figure 4 but based on graphs presented, it is not intuitive that was the case. In addition, $n = 3$ per group does not allow parametric analyses (one cannot guarantee normality).
3. The inconsistency also applies to Ingenuity Pathway Analysis (IPA). Differential expression is now defined as (fold change > 1.2 and an adjusted p -value < 0.05). Then, The DE cutoff for unique networks used for enrichment analysis was $\log_2FC > 0.58$ and raw $p < 0.05$.
4. Sample size is also an issue for other analyses ($n=3$) since two-tailed unpaired Student's t -test or one-way ANOVA are parametric tests. The same applies to the use of mean and SEM. Therefore, non-parametric statistics should be used, and data should be presented as median and IQR.
5. I used critiques 2, 3, and 4 as examples to suggest a careful statistical revision for assuring correctness and consistency.
6. Activated microglia induce "A1 astrocytes" by secreting $IL-1\alpha$, TNF and C1q. This is a key finding from Lidellow's article (*Nature* 541, 481–487 (2017)). Are the same mediators upregulated in MCM? What are the mediators secreted/released by activated endothelial cells (ECM)? The same applies to astrocytic conditioned media (ACM). It would be interesting to see what mediators are upregulated in MCM, ECM, and ACM.

Specific comments:

7. Page 5, lines 107-108. Pathology is specified for The Tg-FDD mouse model ("leads to the vascular accumulation of ADan amyloid") but not for APP/PS1.
8. RIN > 9 was used for NanoString gene expression analysis, and RIN > 8 was RNA-seq. These are very conservative values. Please, explain different values between methods and why using a so high

RIN.

9. DGE and DE are used as abbreviations for differential expression. Please, be consistent.

10. Why did the authors decide to use 15-month-old animals (Tg-FDD and APP/PS1)? Please, justify.

COMMSBIO-21-2235-T: Activated Endothelial Cells Induce a Distinct Type of Astrocytic Reactivity.

Response to Reviewers

We thank the reviewers for their constructive suggestions, which helped us improve our manuscript. Our responses to specific comments are below (original reviewer comments are quoted in *italics*, with our responses in regular font).

Reviewer Comments:

Reviewer 1#

In this manuscript, Taylor et al. describe an astrocyte reactivity state induced by activated endothelial cells that resembles a substate previously described to be induced by activated microglia with some key differences. They show that while similar to microglia-induced astrocyte reactivity, activated endothelial conditioned medium (ECM) is neurotoxic and impairs astrocyte phagocytosis, the MCM and ECM induced astrocytes show differences in gene expression and protein levels. Finally, they show evidence that this ECM-induced subtype is associated with the inflamed vasculature in mouse models of Alzheimer's disease (AD) and in human AD tissue. Overall the findings are exciting and add an additional piece to the puzzle of (reactive) astrocyte heterogeneity.

Major Points

1. However, I wonder whether rather than describing astrocytes as being of type "A1" it might be favorable to refer to them as microglia-induced? It is becoming increasingly clear that there are many more complex substates of reactivity that do not necessarily align with a simple A1/A2 dichotomy. In fact, it looks like based on Fig3 that maybe the authors are dealing with an Interferon-induced astrocyte subtype that has been described recently and is associated with the vasculature as well (PMID 34413515)? It might be worth comparing gene expression profiles of this Interferon-subtype and both MCM- and ECM-induced astrocytes described here? Do endothelial cells release Interferon upon LPS activation?

- As the reviewer suggested, we have rewritten a more balanced introduction and discussion, specifically citing newly published research [1, 2]. We avoided the A1/A2 dichotomy by referring to our activated astrocytes as microglia-induced and endothelial-induced.
- The data presented in figure 3 were obtained by measuring a limited number of transcripts (770) using the NanoString Glia Panel. Therefore, we used the RNAseq data (Figure 4) to compare the gene expression profiles between the newly described interferon-induced astrocyte subtype (PMID 34413515) and the MCM- and ECM-induced astrocytes described in this study. No significant overlap was observed when we compared the gene expression profiles of these astrocyte subtypes. We decided not to include this analysis in the revised manuscript because no overlap was observed, and the interferon-induced subtype was identified using a different approach (scRNAseq data from brain samples instead of bulk RNAseq data from astrocyte cultures). This last point could be a major caveat for proper comparisons of gene expression profiles.

2. Experimentally, my main concern involves the RNA-seq analysis in Fig4. As far as I can tell, rather than using adjusted p-values, the authors used "raw" p-values and many genes highlighted (including all genes in Fig4f) have adjusted p-value cut-offs much higher than applied usually. Ideally, cutoffs should be 0.01-0.05. An FDR of 1 would indicate that 100% of the genes could be false-positives and should therefore not be included in downstream analyses.

We thank the reviewer for pointing out this problem. Due to the large number of multiple testing, we have to adjust (N = 14,205 genes) for the RNA-seq comparison, after multiple test compensation the resulted FDR is inevitably high for the entire genome examined despite large fold changes in the expression levels of many

genes. It is important to mention that the field has started to realize that p-values and the FDR are simply suggestive, not a *bona fide* true or false determinant, the relaxed p-values are necessary for exploring all possible targets for investigation [3, 4]. Therefore, the raw p-value cutoff is only a guide to filter the number of promising genes to a manageable size for experimental validation. It is not solid evidence to show *bona fide* significant difference; the true differences were validated in experimental biological investigation that followed. To avoid confusion, we decided to remove Fig 4f in our revision. A statement clarifying the use of relaxed p values has been included in the manuscript (please refer to Material & Methods section, lines 286 to 289).

3. Another concern is the culture system employed. The inclusion of serum, particularly as high as 10% in astrocyte culture media will irreversibly change the transcriptome and function of these cells. The authors must at least acknowledge this as a caveat of the approach. There is also no purity information provided for these cultures (of any cell type: astrocytes, microglia, or endothelial cells) - as the authors do not complete an endotoxin cleanup following their LPS treatments, any residual LPS carried over in conditioned media experiments could activate contaminating microglia in their astrocyte or endothelial cell cultures - leading to difficult to interpret effects. Given that endothelial TLR4 expression is considerably less (or not present?) when compared to microglia, this purity information becomes integral for all downstream analysis and interpretation. The endothelial response to peripheral inflammation driving astrocyte responses is an exciting and novel mechanism, but a lack of purity data and endotoxin cleanup in media exchange experiments remain a lynchpin of the entire model, I'd think this is required.

- As an internal control, in the laboratory we confirm the purity of our cultures and cell media for every experiment. We apologize for not including this important information in the original submission. We have included the data confirming the purity of all our glial cultures (Sup. Fig. 1a, b). To avoid residual transfer of LPS, for each experiment, cells were carefully but thoroughly washed with warm PBS, transitioned, and maintained in serum-free media to prevent residual serum from transferring over during subsequent experiments. Importantly, we used a Chromogenic Endotoxin Quantification kit to ensure no endotoxin contamination in the conditioned media (below 1 endotoxin unit/ml [5-9]). The endotoxin quantification data have been included as a supplementary figure (Sup. Fig. 1c). Information regarding cell culture purity and endotoxin quantification has been included in the Materials & Methods and Results section (please refer to lines, 191 to 195 and 332 to 336).
- Regarding using fetal bovine serum (FBS) in astrocyte cultures, 10% is a widely accepted and necessary concentration that allows the healthy development of glial cultures [10]. We acknowledge the reviewer's concern regarding irreversible functional and transcriptional changes in astrocytes; therefore, we mentioned this as a possible caveat of the approach in the study (please refer to Discussion section lines, 626 to 630). However, it is important to mention that we were very rigorous in daily assessing our cultures by analyzing the shape, confluency, and overall health of our astrocytes. All cultures that did not meet the criteria of our internal controls were discarded immediately and not used in our experiments. This last statement has been included in the Materials & Methods section (lines 195 to 197).

Minor points:

*4. Fig3: Maybe adding padj-values/FDRs or * to the heatmaps in b and e would make it clearer which genes are significantly different across the two stimulations?*

As the reviewer suggested, asterisks (*) indicating significance have been added to the heatmaps (Fig. 3b and e) to show clearly which genes are significantly different. A description of the analysis performed to determine significant differences has been included in the NanoString analysis section in the Materials & Methods section (lines 262 to 264).

5. Fig4: Is it possible to add a heatmap of all ECM and MCM DEGs to really appreciate the different gene expression profiles?

- As the reviewer suggested, a heatmap of all ECM and MCM DEGs has been included in Sup. Fig. 3.

6. Fig6: Did the authors perform a C3 staining of WT tissue? If so it would be worth including here.

- As the reviewer requested, we have updated Fig. 6 to include GFAP, C3 & Decorin staining of WT tissue.

7. Fig7: Similarly, is there a Decorin staining for non-AD tissue that could be added to this figure?

- As the reviewer suggested, we have updated Fig. 7 to include Decorin staining from non-AD tissue.

Reviewer #2:

This well written and carefully designed manuscript by Taylor et al. reports that injured endothelial cells can induce a unique A1 phenotype in astrocytes, which is characterized by a genetic signature associated with extracellular matrix remodeling factors. A1 astrocytes induced by activated endothelial cells exhibit elevated decorin and are associated with vascular A β deposits in animal and human models of CAA and AD, suggesting a subpopulation of reactive astrocytes, specifically associated with A β -induced endothelial cell dysfunction. This work opens new avenues towards the possibility that multiple different subtypes of C3 positive astrocytes, likely activated by specific cell types, exist. This will allow to further comprehend if (and which) cell-specific molecular players may induce a characteristic pathological phenotype in astrocytes, which may mediate unique types of inflammatory responses in neurodegenerative diseases. Therefore, these findings are novel and informative, and appear to meet the scientific quality level to be suitable for publication in Communications Biology.

A few experimental revisions and text edits are suggested below:

1. It would be important to identify if major differences in cytokines within ECM and MCM may be responsible for the different activation of the astrocytes.

- As the reviewer suggested, to identify if LPS-activated endothelial and microglial cells secrete similar or distinct factors responsible for the activation of astrocytes, we measured the levels of proteins involved in cytokine signaling in ECM and MCM as well as the proteins involved in extracellular matrix composition. As previously reported, we observed an increase in the levels of IL-1 α and TNF- α secreted by activated microglia but not by endothelial cells. No differences were observed in the other proteins measured (MMP2, MMP12, CD93 and GDF-15). These results are shown in Sup. Fig 4 and described in the results section. We also discussed the necessity for further studies to determine the mediators secreted by activated endothelial cells responsible for inducing neurotoxic astrocytes (please refer to materials & methods section lines, 301 to 308 and results lines, 433 to 444).

2. In figures 1-2-5 legends, please add how many cells and images were counted in each experiment. Most graphs represent 3 points. Please clarify what each point stands for.

- As the reviewer requested, we have clarified the figure legends of Fig. 1, 2 and 5 to include how many cells and images were counted in each experiment.

3. Phase contrast images in Fig. 2, panels C, G, N, Q, at the lower magnification are not clear, please replace with higher quality images.

- As the reviewer requested, we have updated Fig. 2, panels C, G, N, Q with higher magnification phase-contrast images of neurons and astrocytes.

4. In Figure 2p, the word "astrocytes" is missing below the petri dish illustration.

- We thank the reviewer for this observation. We have updated Fig. 2p to include the word "astrocytes" below the petri dish illustration.

5. line 124. The antibody is anti-IBA1.

- We have updated line 129 to reflect anti-IBA1.

6. Lines 551-556. Please divide or clarify sentence, which is a bit too long and therefore confusing.

- We thank the reviewer for this point. We have rephrased this sentence to enhance clarity. Please refer to lines 616-619.

7. line 858. In the legends of Figures 6a and 6c, please specify what the pink and yellow arrowheads point to. Moreover, in the legend of Figures 6a, explain the meaning of the white dashed arrows, in the CC GFAP/decorin images.

- We thank the reviewer for this point. As the reviewer requested, we have updated the legend of Fig. 6 to specify yellow arrows, purple arrows and the meaning of the white dashed arrows in the plot profiles in the CC GFAP/Decorin images.

8. Figure 7. In human brains, the authors may consider to co-stain also for C3, GFAP and decorin, to confirm the A1 astrocytic phenotype in decorin positive cells. Additionally, co-staining with C3, GBP2 and decorin would be useful to confirm that these A1 astrocytes are GBP2 negative.

- As the reviewer suggested, we have costained human brain samples for C3, GFAP and Decorin to confirm the A1 astrocytic phenotype in Decorin positive cells. Please refer to Sup. Fig. 6. Additionally, we have included costaining of amyloid (Thio-S), GFAP and GBP2 in human cases, confirming that reactive astrocytes associated with parenchymal amyloid deposits are GBP2 positive and that astrocytes associated with vascular deposits are GBP2 negative. Please refer to Sup. Fig. 7.

9. Figure 7. Please add scale bar in 3D view.

- We thank the reviewer for this suggestion. We have updated Fig. 7 to include a scale bar in 3D view.

Reviewer#3:

The work by Taylor and colleagues investigates whether activated endothelial cells can induce neurotoxic reactive astrocytes. Authors demonstrated that activated endothelial cells induce upregulation of C3 in astrocytes, a hallmark of A1 astrocytes induced by microglial activation. However, they show that C3+ neurotoxic astrocytes induced by activated endothelial cells present a distinct profile than astrocytes activated by microglia. Finally, they demonstrate that endothelial-activated astrocytes are Decorin-positive and seem to be associated with vascular amyloid deposits in mouse models and AD/CAA patients. These findings build upon the idea of astrocyte reactivity heterogeneity and are of interest. The article is well written, and experiments seem of high technical quality. However, some points should be addressed before a potential publication in Communications Biology.

1. The use of A1 to characterize a universal neurotoxic astrocyte phenotype has been widely debated. The recent consensus article proposed to avoid using it (Ref 1 – Escartin et al.). I would advise authors to make a more balanced introduction and discussion about astrocyte heterogeneity.

- Reviewer 1 expressed similar requests, and we have avoided designating A1-astrocytes as a universal neurotoxic phenotype. We have rewritten a more balanced introduction and discussion, specifically citing newly published research on neuroinflammatory astrocyte subtypes, which was not yet published at the time of the initial submission [1, 2]. We have also incorporated reviewer 1's suggestion to avoid the A1/A2 dichotomy by referring to our activated astrocytes as microglia-induced and endothelial-induced.

2. Statistics used for evaluating RNA-seq data are poorly described. For example, differentially expressed genes are defined as (Fold change > 1.5, p<0.05). Are these data unadjusted?

They mention FDR correction in figure 4 but based on graphs presented, it is not intuitive that was the case. In addition, n = 3 per group does not allow parametric analyses (one cannot guarantee normality).

- We thank the reviewer for pointing out this problem. We have included a detailed description of the RNA-seq analysis in the Materials & Methods section (lines 278 to 289). The data were normalized using the TMM (trimmed mean of M values) method. Differential expression analysis was performed using EdgeR (v.3.12.1). Applying the widely accepted R package EdgeR for gene expression comparison is common in the RNA-seq analysis field [11]. It applies a negative binomial model, which fits well with the large range in the RNA-seq count data. No data normality is involved, and data are unadjusted. As in our response to reviewer 1's second question, we did not apply the adjusted p-value (FDR) cutoff to generate DE gene list; instead, for novel pathway exploration purposes, we used the raw p-value.

3. The inconsistency also applies to Ingenuity Pathway Analysis (IPA). Differential expression is now defined as (fold change > 1.2 and an adjusted p-value < 0.05). Then, The DE cutoff for unique networks used for enrichment analysis was log₂FC > 0.58 and raw p<0.05.

- We thank the reviewer for pointing out this problem. Upon reexamining the data and analysis records, we realized that the original description of the DE cutoff was not accurate in the first submission. The same threshold was utilized for all DE, IPA network and enrichment analyses (foldchange > 1.5 and raw p<0.05). We apologize for this mistake, and the correct threshold/cutoff has been included in the revised manuscript (lines 297 to 298).

4. Sample size is also an issue for other analyses (n=3) since two-tailed unpaired Student's t-test or one-way ANOVA are parametric tests. The same applies to the use of mean and SEM. Therefore, non-parametric statistics should be used, and data should be presented as median and IQR.

- We thank the reviewer for this comment. The use of parametric tests has been avoided, and analyses have been performed using nonparametric tests. Sample sizes have been increased to n=4-5 for experiments described in Fig. 1, 2, and 5 & Sup. Fig. 1, 2 and 5 with data presented as median and IQR. The information regarding the statistical analysis has been updated in the Materials & Methods sections (line 318 to 325) and in each figure legend when appropriate.

5. I used critiques 2, 3, and 4 as examples to suggest a careful statistical revision for assuring correctness and consistency.

- Following the reviewer's advice, we have carefully reviewed every statistical analysis performed to ensure correctness and consistency. As in our response to critique four, the information regarding the statistical analysis has been updated in the Materials & Methods sections (line 318 to 325) and in each figure legend when appropriate.

6. *Activated microglia induce "A1 astrocytes" by secreting Il-1 α , TNF and C1q. This is a key finding from Lidellow's article (Nature 541, 481–487 (2017). Are the same mediators upregulated in MCM? What are the mediators secreted/released by activated endothelial cells (ECM)? The same applies to astrocytic conditioned media (ACM). It would be interesting to see what mediators are upregulated in MCM, ECM, and ACM.*

- As the reviewer suggested, to identify if LPS-activated endothelial and microglial cells secrete similar or distinct factors responsible for the activation of astrocytes, we measured in ECM and MCM the levels of proteins involved in cytokine signaling as well as proteins involved in extracellular matrix composition. As previously reported, we observed an increase in the levels of IL-1 α and TNF- α secreted by activated microglia but not by endothelial cells, suggesting that distinct endothelial cell-secreted factors induce this distinctive type of astrocytic activation. No differences were observed in the other proteins measured (MMP2, MMP12, CD93 and GDF-15). These results are shown in Sup. Fig 4 and described in the results section. We also discussed the necessity for further studies to determine the mediators secreted by activated endothelial cells responsible for inducing neurotoxic astrocytes (lines 433 to 444). Regarding the neurotoxic mediators secreted by microglia-induced astrocytes, these mediators were not identified in Lidellow's article in which A1-astrocytes were originally described (Nature 541, 481–487, 2017). During the submission and review of this manuscript, a study was published that suggests that long-chain saturated fatty acids contained in lipoparticles may be the mediators of microglia-induced astrocyte neurotoxicity. However, the specific factors have not yet been identified as it is clear that reducing these lipids does not completely eliminate neurotoxicity, suggesting that future work is needed to discover other astrocyte-derived toxins [12]. In this revised version, we mentioned this new study, and we discussed the necessity for further studies to determine the neurotoxic factors secreted by endothelial-induced astrocytes (lines 361 to 366).

Specific comments:

7. *Page 5, lines 107-108. Pathology is specified for The Tg-FDD mouse model ("leads to the vascular accumulation of ADan amyloid") but not for APP/PS1.*

- As the reviewer suggested, we have specified the pathology for the APP/PS1 model in the Materials & Methods section (lines 112 to 113).

8. RIN > 9 was used for NanoString gene expression analysis, and RIN > 8 was RNA-seq. These are very conservative values. Please, explain different values between methods and why using a so high RIN.

- Based on the guidance of the Center for Medical Genomics Core at Indiana University, the Agilent Bioanalyzer 2100 used for RNA-Sequencing requires a RIN value of 8 or higher when using RNA

extracted from cell culture experiments. All the samples analyzed by RNAseq in this study had good quality RIN values of 9.9-10 using the Purelink RNA mini kit for extraction. This information, including the RIN values of the samples (9.9 – 10), has been included in the Material & Methods section (lines 268 to 276). NanoString technology requires a cutoff of RIN values of >9 for cell culture experiments as their platform operates using a hybridization and capture/reporter probe system for quantifying gene expression, and intact RNA is critical for analysis. As in the RNAseq analysis, all samples utilized for NanoString had a good quality RIN values of 9.9-10. The information clarifying the exact RIN values of the samples used for NanoString has been included in the Materials & Methods section (lines 256 to 257).

9. *DGE and DE are used as abbreviations for differential expression. Please, be consistent.*

- As the reviewer suggested, the abbreviation DGE has been removed and replaced with DE to maintain consistency throughout the manuscript.

10. *Why did the authors decide to use 15-month-old animals (Tg-FDD and APP/PS1)? Please, justify.*

- We apologize for not including this justification in the manuscript. We decided to perform the analysis on 15-month-old animals because there is extensive vascular amyloid accumulation in the Tg-FDD model and parenchymal and vascular amyloid deposition in the APP/PS1 model at this age [13-16]. This justification has been included in the revised version of this manuscript (lines 119 to 121).

References

1. Escartin, C., et al., *Reactive astrocyte nomenclature, definitions, and future directions*. Nature neuroscience, 2021. **24**(3): p. 312-325.
2. Hasel, P., et al., *Neuroinflammatory astrocyte subtypes in the mouse brain*. Nature Neuroscience, 2021. **24**(10): p. 1475-1487.
3. Greenland, S., et al., *Statistical tests, P values, confidence intervals, and power: a guide to misinterpretations*. Eur J Epidemiol, 2016. **31**(4): p. 337-50.
4. Amrhein, V., S. Greenland, and B. McShane, *Scientists rise up against statistical significance*. Nature, 2019. **567**(7748): p. 305-307.
5. Schwarz, H., et al., *Residual endotoxin contaminations in recombinant proteins are sufficient to activate human CD1c+ dendritic cells*. PLoS One, 2014. **9**(12): p. e113840.
6. Gao, H.-M., et al., *Synergistic Dopaminergic Neurotoxicity of the Pesticide Rotenone and Inflammogen Lipopolysaccharide: Relevance to the Etiology of Parkinson's Disease*. The Journal of Neuroscience, 2003. **23**(4): p. 1228.
7. Lee, E.J., et al., *Alpha-synuclein activates microglia by inducing the expressions of matrix metalloproteinases and the subsequent activation of protease-activated receptor-1*. J Immunol, 2010. **185**(1): p. 615-23.
8. Park, J.Y., et al., *Microglial phagocytosis is enhanced by monomeric alpha-synuclein, not aggregated alpha-synuclein: implications for Parkinson's disease*. Glia, 2008. **56**(11): p. 1215-23.
9. Zhang, W., et al., *Aggregated alpha-synuclein activates microglia: a process leading to disease progression in Parkinson's disease*. Faseb j, 2005. **19**(6): p. 533-42.
10. Kaech, S. and G. Banker, *Culturing hippocampal neurons*. Nature Protocols, 2006. **1**(5): p. 2406-2415.

11. Conesa, A., et al., *A survey of best practices for RNA-seq data analysis*. Genome biology, 2016. **17**: p. 13-13.
12. Guttenplan, K.A., et al., *Neurotoxic reactive astrocytes induce cell death via saturated lipids*. Nature, 2021. **599**(7883): p. 102-107.
13. Garcia-Alloza, M., et al., *Characterization of amyloid deposition in the APP^{swe}/PS1^{dE9} mouse model of Alzheimer disease*. Neurobiol Dis, 2006. **24**(3): p. 516-24.
14. Taylor, X., et al., *A1 reactive astrocytes and a loss of TREM2 are associated with an early stage of pathology in a mouse model of cerebral amyloid angiopathy*. Journal of Neuroinflammation, 2020. **17**(1): p. 223.
15. Cisternas, P., et al., *Vascular amyloid accumulation alters the gabaergic synapse and induces hyperactivity in a model of cerebral amyloid angiopathy*. Aging Cell, 2020. **19**(10): p. e13233.
16. Szu, J.I. and A. Obenaus, *Cerebrovascular phenotypes in mouse models of Alzheimer's disease*. J Cereb Blood Flow Metab, 2021. **41**(8): p. 1821-1841.

REVIEWERS' COMMENTS:

Reviewer #1 (Remarks to the Author):

The authors have made extensive edits and additions to their original manuscript and have addressed the majority of the concerns raised in the original round of reviews.

In particular I appreciate the additions to the narrative that have cleared up much of the initial confusion in the first submission.

I would also like to commend the authors for only using RNA samples of extremely high quality (RIN 8, and 9.9-10). While this was raised as a concern by another reviewer I think the authors have made the correct decision in being so stringent.

I must however question once more, the use of serum in astrocyte cultures. It has now been a decade since serum-free culture methods have been published (Foo et al., Neuron 2011), and the statement that '10% is standard in the field' is simply unacceptable. The astrocytes pictured in these figures are not healthy, nor physiologically normal. Such a flattened, fibroblast-like morphology in no way recapitulates that of astrocytes in vivo, or in serum-free media. Equally, several studies have shown that serum inclusion changes expression levels of hundreds of genes, and drastically alters protein levels and function. The authors must clearly state the caveats of using such serum-containing methods, otherwise continued use of dated and inadequate methods will continue to perpetuate in the field. While there are instances where the results generated using these methods can be recapitulated in vivo, the vast majority have reported artifacts that cannot be replicated, validated, and have moved the field away from biologically relevant discoveries and use of appropriate models.

Reviewer #2 (Remarks to the Author):

The authors addressed the majority of the experimental revisions requested and text edits that we suggested, therefore the manuscript is suitable for publication.

Minor comments:

-The main mediators in ECM and MCM initiating the different activation of astrocytes have not been fully elucidated, and must further investigated, as now mentioned in the manuscript. The authors only analyzed the levels of IL-1 α and TNF α , typically characterizing microglia activation, and MMP2, MMP12, GDF-15 and CD93, which are proteins involved in the extracellular matrix signaling. It would be important to include in the discussion, as a limitation of this study, that only a small number of possible mediators have been analyzed in this study.

- In Fig. 1 and 5 legends, the authors clarified how many cells and images were counted in each experiment, but not in Fig.2.

Reviewer #3 (Remarks to the Author):

The authors addressed all my primary concerns. However, I have a few final considerations.

1) A1 phenotype is still highly used by the authors. Therefore, I suggest authors use it always together with "microglia-induced".

2) The exploratory nature of the transcriptomic analyses should be clearly stated since sample size is very small ($n=3-4$ for omics), and analyses were not corrected by FDR. For example, figures 3 and 4 are entirely exploratory with uncorrected findings. The figure legends should have the information that the data are uncorrected.

COMMSBIO-21-2235A: Activated Endothelial Cells Induce a Distinct Type of Astrocytic Reactivity.

Response to Reviewers

We thank the reviewers for their constructive suggestions, which helped us improve our manuscript. Our responses to specific comments are below (original reviewer comments are quoted in *italics*, with our responses in regular font).

Reviewer Comments:

Reviewer 1#

The authors have made extensive edits and additions to their original manuscript and have addressed the majority of the concerns raised in the original round of reviews. In particular I appreciate the additions to the narrative that have cleared up much of the initial confusion in the first submission. I would also like to commend the authors for only using RNA samples of extremely high quality (RIN 8, and 9.9-10). While this was raised as a concern by another reviewer I think the authors have made the correct decision in being so stringent. I must however question once more, the use of serum in astrocyte cultures. It has now been a decade since serum-free culture methods have been published (Foo et al., Neuron 2011), and the statement that '10% is standard in the field' is simply unacceptable. The astrocytes pictured in these figures are not healthy, nor physiologically normal. Such a flattened, fibroblast-like morphology in no way recapitulates that of astrocytes in vivo, or in serum-free media. Equally, several studies have shown that serum inclusion changes expression levels of hundreds of genes, and drastically alters protein levels and function. The authors must clearly state the caveats of using such serum-containing methods, otherwise continued use of dated and inadequate methods will continue to perpetuate in the field. While there are instances where the results generated using these methods can be recapitulated in vivo, the vast majority have reported artifacts that cannot be replicated, validated, and have moved the field away from biologically relevant discoveries and use of appropriate models.

- As the reviewer suggested, we discussed in the manuscript the caveat of working with serum-containing astrocytes and the necessity of further studies utilizing serum-free methods and brain samples. We have included Foo et al., Neuron 2011 as a reference after this statement. Please refer to discussion section, lines 407– 413.

Reviewer 2#

The authors addressed the majority of the experimental revisions requested and text edits that we suggested, therefore the manuscript is suitable for publication.

Minor comments:

The main mediators in ECM and MCM initiating the different activation of astrocytes have not been fully elucidated, and must further investigated, as now mentioned in the manuscript. The authors only analyzed the levels of IL-1 α and TNF α , typically characterizing microglia activation, and MMP2, MMP12, GDF-15 and CD93, which are proteins involved in the

extracellular matrix signaling. It would be important to include in the discussion, as a limitation of this study, that only a small number of possible mediators have been analyzed in this study.

- As the reviewer suggested, we included in the discussion a statement regarding the small number of mediators analyzed and the necessity of further studies to determine the exact factor secreted by endothelial cells responsible for activating astrocytes into a C3+Decorin+ phenotype. Please refer to discussion section, lines 312 – 315.

- In Fig. 1 and 5 legends, the authors clarified how many cells and images were counted in each experiment, but not in Fig.2.

- As the reviewer requested, we have included the number of cells analyzed in Figure legend 2.

Reviewer 3#

The authors addressed all my primary concerns. However, I have a few final considerations.

1) A1 phenotype is still highly used by the authors. Therefore, I suggest authors use it always together with "microglia-induced".

- As the reviewer suggested, we always used “microglia-induced” together with “A1 phenotype”.

2) The exploratory nature of the transcriptomic analyses should be clearly stated since sample size is very small (n=3-4 for omics), and analyses were not corrected by FDR. For example, figures 3 and 4 are entirely exploratory with uncorrected findings. The figure legends should have the information that the data are uncorrected.

- As the reviewer suggested, we have included in Figure legends 3 and 4 the following sentence: “Due to the small sample size and exploratory purpose, statistical significance is reported by raw p values, and is not corrected by FDR.”